# Multi-scale Measurements of Mesospheric Aerosols and Electrons During the MAXIDUSTY Campaign

Tarjei Antonsen[1], Ove Havnes[1], and Andres Spicher[2]

[1]Department of Physics and Technology, UiT- The Arctic University of Norway
[2]Department of Physics, University of Oslo

**Correspondence:** T. Antonsen (tarjei.antonsen@uit.no)

**Abstract.** We present in-situ measurements of small scale fluctuations in aerosol populations as recorded through a mesospheric cloud system from the Faraday cups DUSTY and MUDD during on the MAXIDUSTY-1 and 1B sounding rocket payloads launched in the summer of 2016. Two mechanically identical DUSTY probes mounted with an inter-spacing of $\sim 10$ cm recorded very different currents, with strong spin modulation, in certain regions of the cloud system. A comparison to auxiliary measurement show similar tendencies in the MUDD data. Fluctuations in the electron density are found to be generally anti-correlated to the negative aerosol charge density on all length scales, however, in certain smaller regions the correlation turns positive. We have also compared the spectral properties of the dust fluctuations, as extracted by wavelet analysis, to PMSE strength. In this analysis, we find a relatively good agreement between the power spectral density (PSD) at the radar Bragg scale inside the cloud system, however the PMSE edge is not well represented by the PSD. A comparison of proxies for PMSE strength, constructed from a combination of derived dusty plasma parameters, show that no simple proxy can reproduce PMSE strength well throughout the cloud system. Edge effects are especially poorly represented by the proxies addressed here.

## 1 Introduction

The terrestrial mesosphere, situated at $\sim 50 - 100$ km, contains the ambient prerequisites to house a number of different types of nanoparticles. From sub-nanometer sized meteoric smoke particles (MSP) coagulated from ablation vapors of meteors, to ice particles with radii of several tens of nanometers, aerosols in this region vary greatly in composition and size. Such variation consequently makes mesospheric ice and dust particles important in many physical and chemical processes in the atmosphere. The summer mesosphere is particularly interesting in the study of ice and dust particles due to extremely low temperatures, often $\lesssim 120$ K at the mesopause (Lübken, 1999; Gerding et al., 2016), which allows for nucleation of ice into aerosols of sizes up to several tens of nanometers. The summer mesopause region, located between $\sim 80$ and 90 km, is the only region with consistently low temperatures for ice particles to form regularly. Ice particles of sizes $\gtrsim 10$ nm can scatter light effectively and consequently give rise to the phenomenon called noctilucent clouds (NLC). Subvisual particles can also produce coherent radar echoes at frequencies between some tens of MHz and $\sim 1$ GHz, by reducing the electron diffusivity such that gradients in electron density can persist for long time periods and produce radar backscatter at the radar Bragg-scales. Such echoes

are called Polar Mesospheric Summer Echoes (PMSE: see e.g. Rapp and Lübken (2004), Mesospheric ice: see e.g. Rapp and Thomas (2006) for comprehensive reviews).

Due to the height range of the mesosphere, it is unaccessible for balloons, and rocket probes are the only means of in situ observation. Remote measurements are readily carried out from ground and satellites, but some ground measurements are contingent on lower atmosphere conditions while satellite measurements depend on orbit type. For a full characterization of the dusty plasma in the mesopause region, conventional payloads for this purpose must contain probes for detection of electrons, ions and dust and ice particles. Conventional Langmuir probes are convenient in measuring ambient plasma densities, however, different problems may arise in the calibration of these (Bekkeng et al., 2013; Havnes et al., 2011b). Dust particle measurements are often carried out with Faraday buckets, which are electrostatic probes designed to separate charged particles from ambient ions and electrons (see e.g. Havnes et al. (1996); Gelinas et al. (1998)). As with Langmuir probes, calibration of Faraday buckets is a possible issue. Further problems connected to particle dynamics are also typical for mesospheric rocket probes, and modeling of neutral gas flow and electric field structure is often required. Studies of the cut-off of observable sizes in Faraday buckets have shown that at altitudes around 85 km, MSPs with radii $\lesssim 1 - 2$ nm are swept away in the shock in front of the probes, while the cut-off radius for ice particles is somewhat higher (Hedin et al., 2007; Antonsen and Havnes, 2015). Furthermore, secondary charging effects must be considered to correctly interpret measured currents (Havnes and Næsheim (2007); Kassa et al. (2012), Havnes et al. (2019) – this issue).

**Small-scale measurements in the mesopause region**

Observations of mesospheric dust structures on the smallest scales possible are especially interesting in explaining UHF PMSE, diffusion processes and size sorting among other phenomena in the mesopause region. These phenomena are not particularly well understood, and small scale density variations of aerosols and their connection to neutral turbulence and electron density still require substantial observational and theoretical work to be fully comprehended. Few previous studies have emphasized on simultaneous measurements of dust and electron populations. Rapp et al. (2003a) studied the simultaneous variation of electrons and aerosols, and the spectral properties of their fluctuations. They found that there was a general anti-correlation between electrons and charged particles, and that the connection to neutral turbulence was clear. The anti-correlation has been observed on large scales since the early days of mesospheric rocket studies (see e.g. Pedersen et al. (1970)), but its precence on the smallest scales is not the general rule. Lie-Svendsen et al. (2003) showed that a correlation between ions and electrons, thus complicating the relationship with dust particles, can be positive in regions of high aerosol evaporation and large particles. Strelnikov et al. (2009) studied the connection to neutral turbulence, substantiating the connection between mesospheric dust and VHF PMSE.

In this work we present the measurements from the MAXIDUSTY campaign, with special emphasis on the MAXIDUSTY-1B payload launched from Andøya Space Center, 8th of July 2016. The experimental framework utilized in the present paper consists of two front-mounted mechanically and electrically identical DUSTY Faraday buckets with an interspacing of $\sim 10$ cm, three front-mounted modified Faraday cups of the type MUDD (see Havnes et al. (2014); Antonsen and Havnes (2015); Antonsen et al. (2017)) with similar interspacing, and electron measurements with boom-mounted needle Langmuir probes

(U. of Oslo). The DUSTY probe (see Havnes et al. (1996)) can yield absolute dust charge number density, and the setup on MAXIDUSTY-1B is intended to study horizontal density variations of dust on very short length scales down to the typical sampling resolution of rocket Faraday cups and Langmuir probes which. With this framework we aim to resolve structures in the aerosol population on both horizontal and vertical scales of 10 cm, which can be used to infer blob or hole structures in the dusty plasma on these scales. This knowledge and similar setups can be used in studies of e.g. UHF PMSE which occur during scattering at Bragg-lengths at scales comparable to the smallest scales observed here. A key question addressed here, is how well the aerosol and electron populations correlate. This question have been addressed in several earlier works for large-scale structures – e.g. electron bite-outs associated with PMSE – but a thorough inquiry on the electron-aerosol relationship on smaller scales is to our knowledge seldomly done. With additional information about the number density and size of the aerosols (method described in the companion paper Havnes et al. (2019)), our dataset is well suited to inquire about the relationship between PMSE and charged particles; especially on the role of charged aerosols in PMSE formation. We combine the information on the charged species to test the notion that PMSE strength can be predicted by proxy from fundamental dusty plasma parameters. We have also looked into the spectral properties of the aerosol and electron measurements. A by-product of our investigations is a confirmation of that the aerodynamic environment around typical rocket payloads can dictate the movement of aerosols, and small-scale measurements can be heavily affected by such adverse effects.

We find that the DUSTY probes recorded very different currents in certain parts of the dust layer, while almost identical currents in other parts of the layer, suggesting that the assumption of homogeneity of the dust and/or flow structure across the payload top deck is not always valid. The MUDD probes confirm the DUSTY measurements and display a similar difference between probe currents. A comparison to electron measurements shows that the correlation is generally clearly negative between dust number densities and electron densities, but in some regions of the cloud system the correlation is more variable and not as unambiguous. Results from the spectral analysis of fluctuations in the aerosol population are discussed in the framework of simultaneous PMSE observations done with the IAP MAARSY radar. Lastly, we discuss the applicability and validity of simple proxies composed of the dusty plasma parameters in predicting PMSE strength and shape.

## 2 The DUSTY Faraday Bucket

The schematics of the DUSTY probe are shown in Fig. 1, and the principle of current generation in DUSTY is shown in Fig. 2. The top grid is set to payload potential and is intended to shield neighboring probes from internal electric fields. The grid G1 is biased at +6.2 V in order to deflect ambient ions and absorb ambient thermal electrons. The G2-grid was originally intended to absorb secondary electrons ejected from the bottomplate (BP), to correct for this loss in the derivation of the dust charge number density (Havnes et al., 1996; Havnes and Næsheim, 2007). However, as justified by observations and theoretical considerations, the secondary production at G2 is the dominating secondary charge source and no detectable secondary charge production takes place at the bottom plate. This finding facilitates the utilization of DUSTY to measure dust sizes and absolute number densities of dust particles (Havnes et al., this issue) .

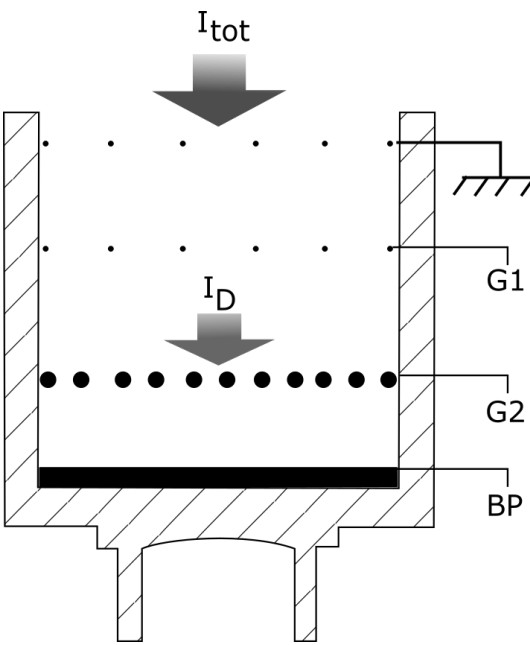

**Figure 1.** Cross section of the DUSTY probe. The upper grid is payload ground intended to shield neighboring probes from E-fields. The Grids G1 and G2 and the bottom plate (BP) have potentials optimized to shield ambient plasma and detect mesospheric dust and ice particles. The wire thickness is exaggerated there for convenience, and we also note that the G2 wires are thicker that the G1 and shielding grid wires.

As indicated above, it has been found that particles of sizes $\lesssim 1-2$ nm are heavily affected by air flow around the probe in the mesopause region (Hedin et al., 2007; Antonsen and Havnes, 2015; Asmus et al., 2017). In the following, we will therefore assume that these particles contribute little to the total dust number density. Such an assumption can be further justified by the notion that very small particles can be neutralized effectively by photo-detachment during sunlit conditions. The dust currents

5  to grid G2 and BP can then be expressed as:

$$I_{G2} = \sigma I_D + I_{sec} \tag{1}$$
$$I_{BP} = (1-\sigma)I_D - I_{sec} \tag{2}$$

where $I_D$ is the current between G1 and G2 as shown in Fig. 1, and $\sigma = 0.28$ is the effective area factor of G2. Note that we here have neglected the secondary contribution from G1 and G0 since their cross section is much smaller than that of G2

10  ($\sim 18\%$ of the area). Since the secondary producing part of the grid is even smaller, the total contribution from the upper grid is small (only about 5%). In a full treatment, this is taken into account, but the contribution to the total derived charge number density is relatively small. We can furthermore relate $I_D$ to the dust charge density $N_d Z_d$ according to:

$$I_D = (1-\alpha)N_d Z_d e v_R \pi R_p^2 \cos\gamma \tag{3}$$

where $v_R$ is the rocket speed, $e = 1.6 \cdot 10^{-19}$ C the elementary charge, $R_p$ is the probe radius, $\gamma$ is the coning angle and $\alpha = 0.08$ is the fraction of the probe area covered/shadowed by G1 and G0. Here we have neglected any secondary production of charge at G1, and the secondary contribution to the currents is denoted by $I_{sec}$. From laboratory studies is has been found that the net contribution of this term is positive during exposure to ice particles less than a few minutes, meaning that dust particles

5    rub off electrons from grid wires in a triboelectric fashion, as illustrated in Fig. 2 (Tomsic, 2001; Havnes and Næsheim, 2007; Kassa et al., 2012). This effect requires a grazing angle of around 70 to 75 degrees to be maximized, if the particles are pure ice (Tomsic, 2001). We also note that combining the equation yields $I_D = I_{G2} + I_{BP}$, as expected.

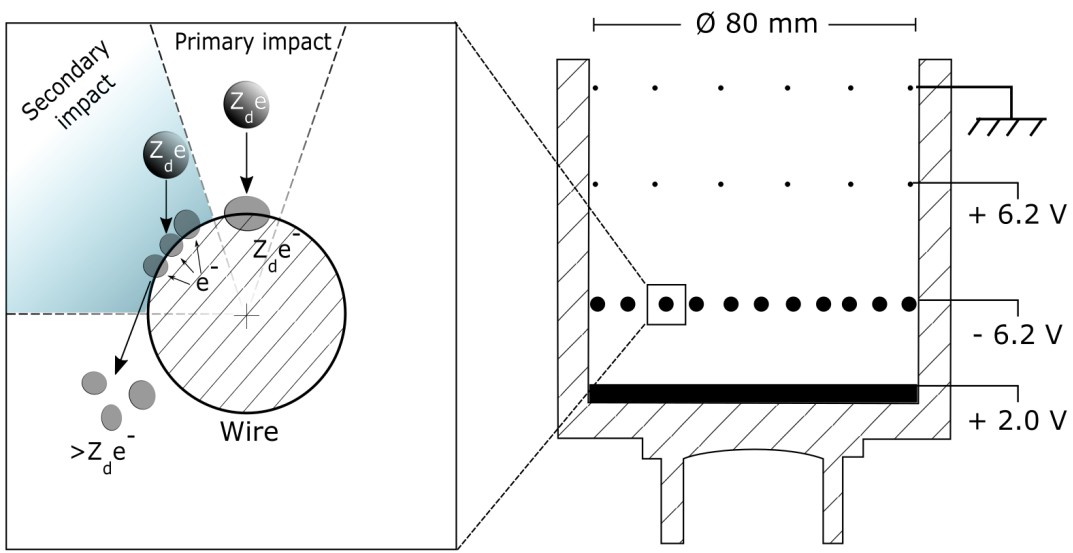

**Figure 2.** Principle sketch of large, order of 10 nm, particles entering DUSTY as launched on the MAXIDUSTY payloads. The mechanism can be described as follows: (1) A large particle deposits its charge in a primary impact and is partly fragmented, (2) If the impact is grazing, fragments can steal electrons from the grid wire. For large particles, the fragments tend to take away more electrons from the wires than the incoming charge and the net current to G2 becomes positive. For small particles, the primary charge is usually larger than the fragment current, and the net current to G2 thus becomes negative. In both cases, the bottom plate current becomes negative. We note that the secondary impact area region is exaggerated here; the true secondary charge producing area is $\gtrsim 20\%$.

Figure 3 shows the mechanical layout of the topdeck on the MXD-1B payload. The layout was similar to the MXD-1 topdeck layout, only with one DUSTY probe replacing the miniMASS aerosol spectrometer (CU Boulder). In total five dust detectors

10    were included on the second flight, of which three were of the type MUltiple Dust Detector (MUDD) and two were identical DUSTY probes. The topdeck also contained sun sensors (denoted DSS in the figure) for orientation measurements, and the Identification of the COntent of NLC particles (ICON) neutral mass spectrometer (see Havnes et al. (2015)). Measurements of electron density where made by Faraday rotation (TU Graz) and multi-needle Langmuir probes (mNLP, U. of Oslo). A Positive Ion Probe (PIP) and a Capacitance probe were mounted on booms (TU Graz). Due to the high sampling rate of the mNLP-

instrument, its data is best suitable for comparison of simultaneous small scale fluctuations in aerosol and electron populations and it will therefore be utilized in the comparison between aerosol and electron fluctuations below.

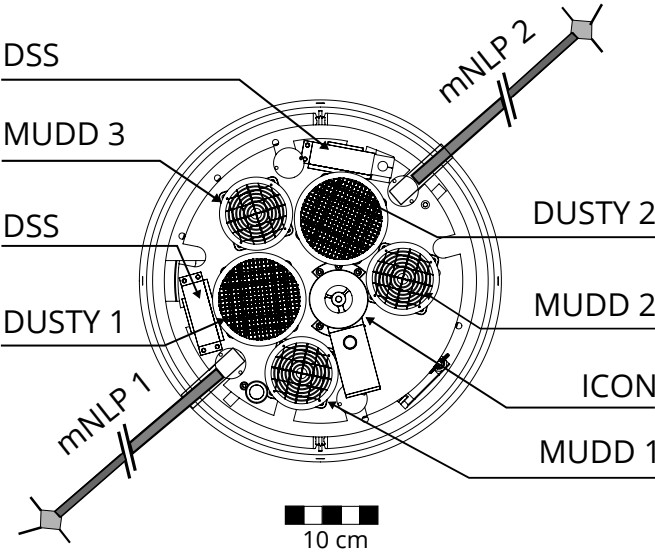

**Figure 3.** Layout of the topdeck and mNLP booms on the MXD-1B payload. The two identical DUSTY probes have a distance between them of  10 cm from center to center. The length of the booms was $\sim 60$ cm, with the aim to minimize aerodynamic and electric adverse effects from the main payload structure.

## 3 DUSTY measurements from the MXD-1B launch

As this work focuses on small-scale measurements of fluctuations in the mesospheric dusty plasma, we use the MXD-1B flight in a case study as it had the dual DUSTY configuration introduced above. DUSTY data from the first flight (MXD-1) gives the basis for the two recent papers of Havnes et al. (2018) and Havnes et al. (2018, this issue), and in this work we also briefly discuss measurements from that payload. The MXD-1B payload was launched from Andøya Space Center (69.29°N, 16.02°E) at 13:01 UT on July 8, 2016. Simultaneous PMSE measurements done with the MAARSY 53.5 MHz VHF radar, recorded an unusually strong PMSE stretching from $\sim 84$ to $\sim 88$ km in altitude. Due to visibility issues, NLC observation by lidar was unavailable at the time of launch.

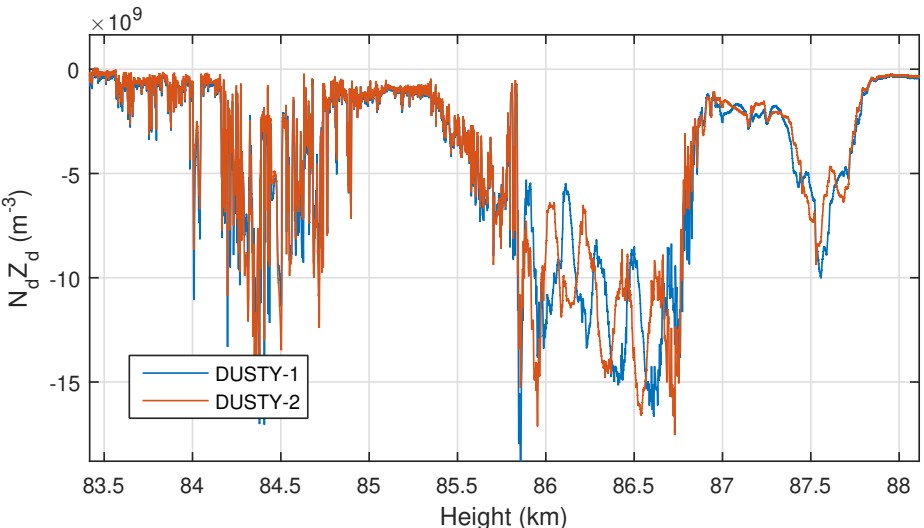

**Figure 4.** Dust charge number density for the identical DUSTY probes launched on the MXD-1B payload on the 8th of July 2016.

A main motivation behind launching two identical probes with a short distance between them, is to characterize the two dimensional structure of dust clumps and holes throughout the cloud region on the shortest scales – i.e. scales on which UHF PMSE are produced. If the dust clumps are made up of dust particles which are large enough to be unaffected by the airflow around the payload, and that the DUSTY probes have no leakage of ambient plasma, the currents measured by DUSTY-1 and DUSTY-2 should be identical. Discrepancies between probe signals imply that aerodynamic effects or other adverse effects are important. We see from the dust charge density derived from the two DUSTY probes in Fig. 4, however, that such a simple similarity is not the case at all heights. Taking the ratio between probe BP currents, $I_{BP,1}/I_{BP,2}$, yields a ratio near unity in the lower part of the cloud system, but from the middle of the cloud the ratio deviates from 1. Between 86 and 86.8 km the difference between the two probes is particularly large. Figure 5 shows the onset of the first disagreement region which starts at $\sim 85.85$ km. Below this altitude DUSTY-1 and DUSTY-2 measurements follow each other closely, but at altitudes above, the

currents are strongly influenced by the rotation of the payload and we see that the two probes here vary roughly in antiphase. The phase difference is very close to the 125° azimuth angle difference between the probes on the front deck (see Fig. 3).

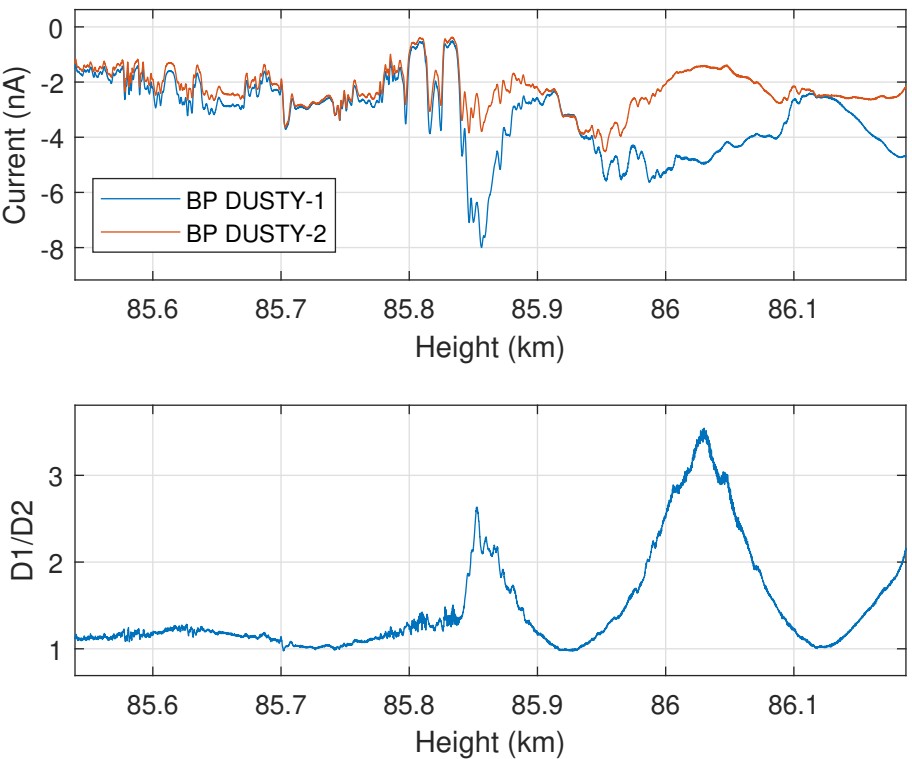

**Figure 5.** Magnification of the immediate region around the onset height ($\sim 85.85$ km) of the large disagreement between the DUSTY probes. Above this height, the ratio of the two DUSTY currents becomes heavily modulated with a characteristic oscillation at the payload spin frequency.

Figure 6 shows the BP currents over approximately two rotation periods below the onset altitude. A weak modulation of the ratio $I_{BP,1}/I_{BP,2}$ with payload rotation is present ($\approx 3.8$ Hz), but the agreement is very good down to the smallest scales $\lesssim 1$ m.

It seems obvious that the main factor in the disagreement between the probes has to be the air stream around the payload which can affect dust particles, particularly the very small ones below one or two nanometer which can be totally swept away from the probes. However, also the somewhat larger dust particles will be affected by the air stream and have their velocity direction affected. If the payload had no coning, so that the payload velocity is directed along its axis, we would expect no change due to rotation unless a strong external wind, at a large angle to the payload axis, could introduce some asymmetry in the air stream. For mesospheric rockets with apogees $\lesssim 140$ km, we expect an angle between payload velocity and axis of

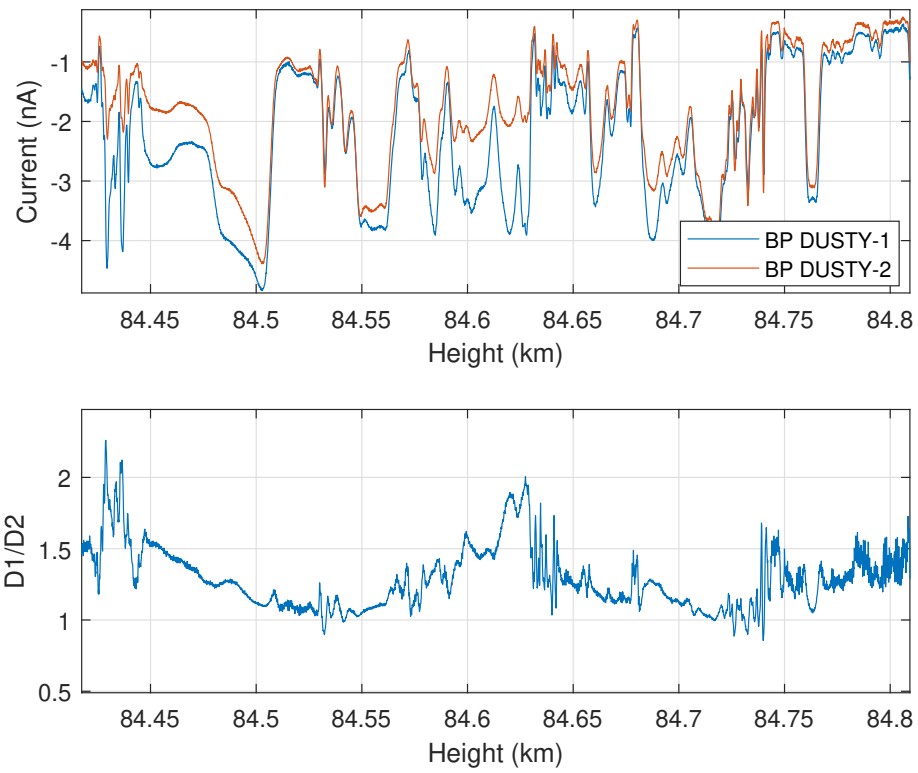

**Figure 6.** Magnification of region with relatively strong probe currents below disagreement onset. A generally good agreement is found down to the lowest height scales ($\sim 10$ cm), which is justified by the $D1/D2$-ratio being near unity.

8-10 degrees throughout the cloud region, which was confirmed by magnetometer orientation data. Also, the asymmetry of the instruments on the front deck could lead to an asymmetry of the air stream even with zero coning. Additionally, ambient plasma may affect recorded currents if the payload becomes substantially charged. The complete characterization of the aerodynamic environment around the supersonic payload flying through a mesospheric dusty plasma is a phenomenal problem to attack, and will not be the main focus of this work. Nevertheless, it is very probable that findings about adverse effects related to aerodynamics and payload charging on the MXD payloads can be transferred with some generality to similar datasets.

Moreover, we have a new tool to further substantiate the claim of small dust particles. By iterating the dusty plasma equations for charge balance and equilibrium between charge states simultaneously (Havnes et al. 2019, this issue), it is possible to calculate the mean dust radius with very good height resolution in a layer of dust from DUSTY-currents. By assuming quasi-neutrality, a secondary charging probability propotional to cross-section and that the aerosols are charged only by electron attachment and polarization effects, one can obtain an equilibrium solution for mean aerosol size, density and charge. In Fig. 7 we show the result of such a calculation for the MXD-1B. The thin and high peaks occurring at certain heights are regions

where the equation for radius approaches 1/0 in the iteration. Such cases usually occur around cloud edges, so the method is more reliable inside clouds. In general, the particle sizes are relatively small throughout the cloud system and only passes 20 nm below $\sim 84$ km. The results above 88 km may be difficult to interpret due to the very low amount of dust there. Since DUSTY currents are directly proportional to the charge number density of dust particles, the iteration scheme mentioned above can be used to obtain the total density of aerosols, $N_d$, also seen in Fig. 7. In the further discussion of how DUSTY currents relate to electron density, we note from the figure that the number density of aerosols is $\sim 10^8 - 10^{10}$ m$^{-3}$. Compared to electron density measurements from Faraday rotation (Friedrich, M., private communication, 2018), this is one to three order of magnitude lower than $N_e$ throughout the layer, which justifies that we can utilize theory on PMSE reflectivity which is valid for low values of $\Lambda = N_d Z_d / N_e$ when investigating the relationship of aerosols to PMSE strength below.

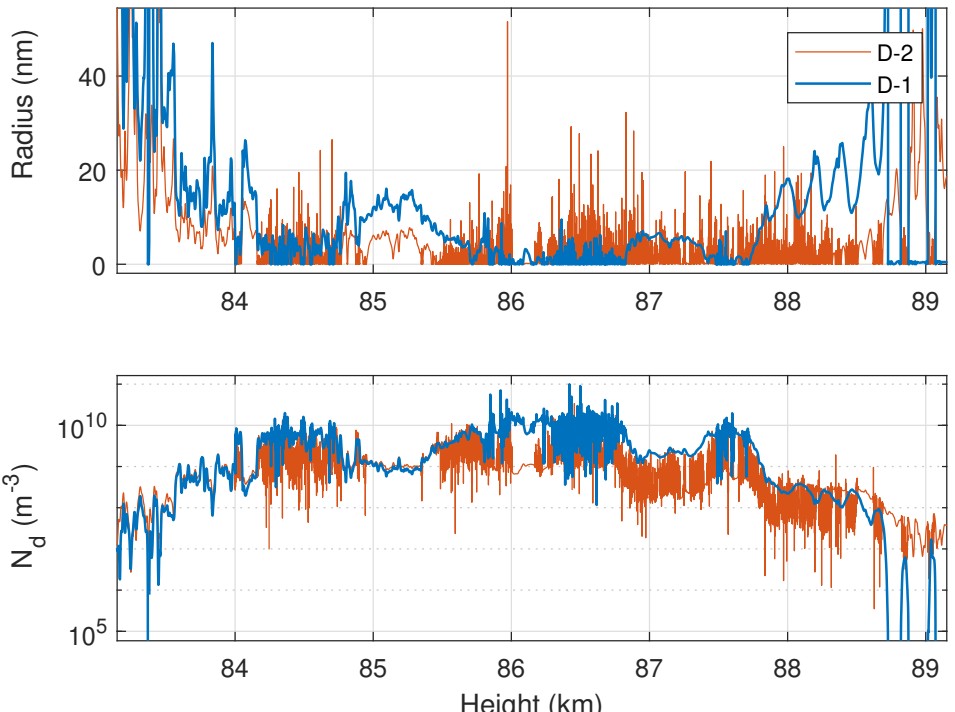

**Figure 7.** Particle radius and number densities derived from DUSTY-1 (blue) and DUSTY-2 (red) data through the method introduced by Havnes et al. (2018, this issue). The sizes are generally small and densities are generally high compared to earlier flights and values usually found in lidar studies.

## 4 Comparison to auxiliary mesurements

The DUSTY and MUDD Faraday cups alone present a vast amount of data; in total 27 electrometers for the two flights combined. In the present work we use the current recorded on the bottom plate and lowest grid (G2) channels on DUSTY, since the dust number density can be directly inferred from these measurements. We neglect the effect of secondary effects
from the topmost grid (G1), and the currents for these are dominated by the absorbed ambient plasma particles and is thus difficult to use for dust measurements. It is possible however to use the auxiliary grids in a calibration of electron probes (see e.g. Havnes et al. (2011a)), but as the Langmuir probes on MXD operated in a different regime (OML), it is complicated to do in-flight calibration. The MUDD measurements are only used here to support the DUSTY measurements. Due to a more complex geometry and electronic settings (with rapidly switching potentials), it is more complex to obtain the number density
of charged aerosols directly, and DUSTY is best suited for that purpose. The same limitations in using the G1 currents on DUSTY also applies for MUDD. Antonsen et al. (2017) presented a detailed description on how to utilize the MUDD currents to obtain the size distribution of meteoric smoke particles, and used that method on the MXD-1 and 1B flight data. The companion paper of Havnes et al. (2019) studies the utilization of DUSTY to obtain information on aerosol number density, where the current paper observes the aerosol-electron connection from a somewhat greater distance.
In the current section we describe the accompanying measurements carried out by the on-board MUDD and Langmuir probes. The MUDD probes are used here first and foremost as a control of the DUSTY measurements, as we expect a certain connection between the two.

### 4.1 Secondary Impact Currents: MUDD measurements

As a control of the DUSTY measurements we address the similarity of the MUDD measurements to the measurements from the
DUSTY probes (see Havnes et al. (2014); Antonsen and Havnes (2015) for a technical description of MUDD.). The principal difference between a DUSTY and a MUDD probe is that in the latter, the G2 grid is replaced with an opaque grid consisting of inclined concentric rings to ensure that all particles hit a ring. The principle is that the secondary current should become large compared to DUSTY, since in MUDD the area producing secondary charging now is equal to the full opening of DUSTY (i.e. $\rho = 1$ in eqs. 1 and 2). On the MXD-1B payload, three MUDD probes were mounted on the topdeck with an azimuthal angle
of $\sim 120°$ between them. For comparison to DUSTY, we look at the currents from the MUDD-1 and MUDD-3 probes since these had observation modes with attracting potentials to ensure that even the smallest impact fragments were measured. A comparison of the bottom plate current of MUDD to charge number density derived from DUSTY is shown in Fig. 8. There is a good agreement between the two throughout the cloud. In the region starting at $\sim 85.9$ km, the disagreement between the MUDD probes is even more pronounced than for the two DUSTY probes. The phase difference between peaks in this region
is also here consistent with the azimuthal difference between the probes. The MUDD currents differ from DUSTY above $\sim 88$ km. In this region, the MUDD currents are stronger than below the lower layer dust cloud, as opposed to DUSTY where the topside currents are effectively zero. In Fig.9 we show the correlation between MUDD-1 and MUDD-3 total current. These two probes had channels which could measure the total current of incoming charged aerosols, and all their charged fragments

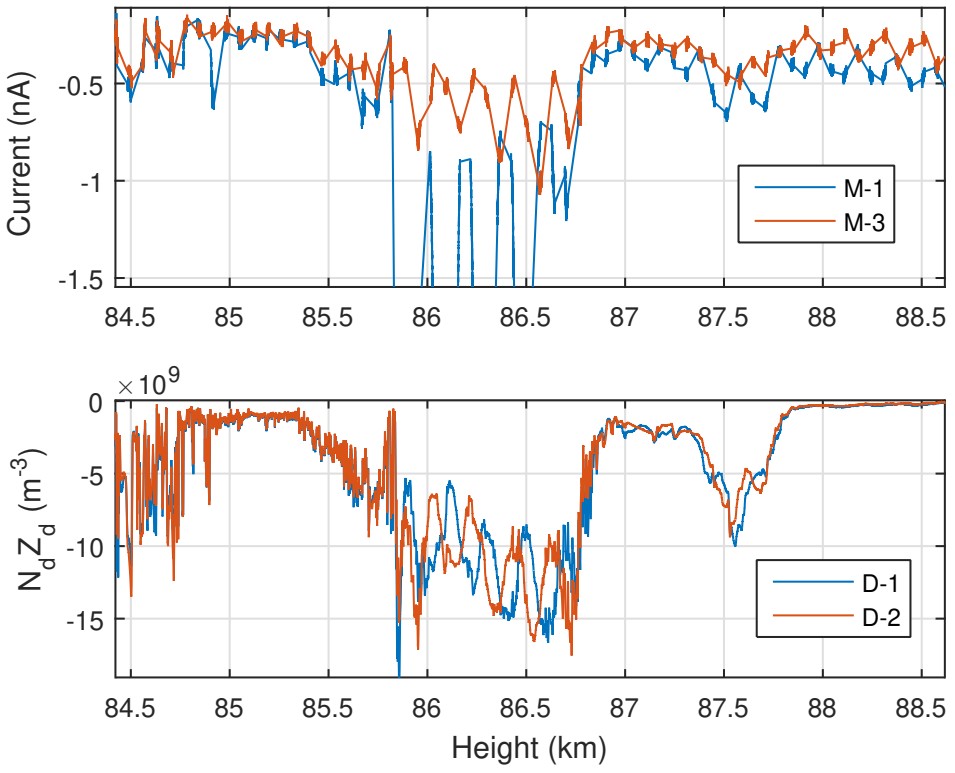

**Figure 8.** Comparison of MUDD-1 and MUDD-3 currents (top) and DUSTY-1 and DUSTY-2 charge number densities (bottom). Both probe pairs display the same heavily spin modulated feature at $\sim 86$ km, suggesting the presence of very small dust particles.

produced on impact with the probe. Such a measurement can be directly related to DUSTY by assuming the same secondary charging efficiency of the probes, and can accordingly be compared to DUSTY without any particular loss of generality. Due to the angle between the probes of $120°$, if the currents were completely dominated by payload rotation, the correlation would be negative. Consequently, if the angle between the probes were $180°$ the correlation would be $-1$ in such a situation. At the bottom of the cloud at $\sim 83$ km, the correlation rises to almost unity, indicating that large particles dominate the currents. The correlation analysis also reveals that there is a strong variation in the relationship between the MUDD-1 and MUDD-3 currents above this region. Since this analysis is unaffected by spin modulation, it is possible to infer structures which normally would be difficult to separate from the background. Interestingly, two regions above 90 km, one centered at $\sim 91$ km and one centered at $\sim 93$ km, show a tendency of a weaker correlation than the expected value which is close to unity. This might suggest that there are populations of very small particles which control the electrons and thus the electron leakage current to MUDD at these altitudes. If the payload potential is negligible, we would expect the correlation to very close to unity at these heights.

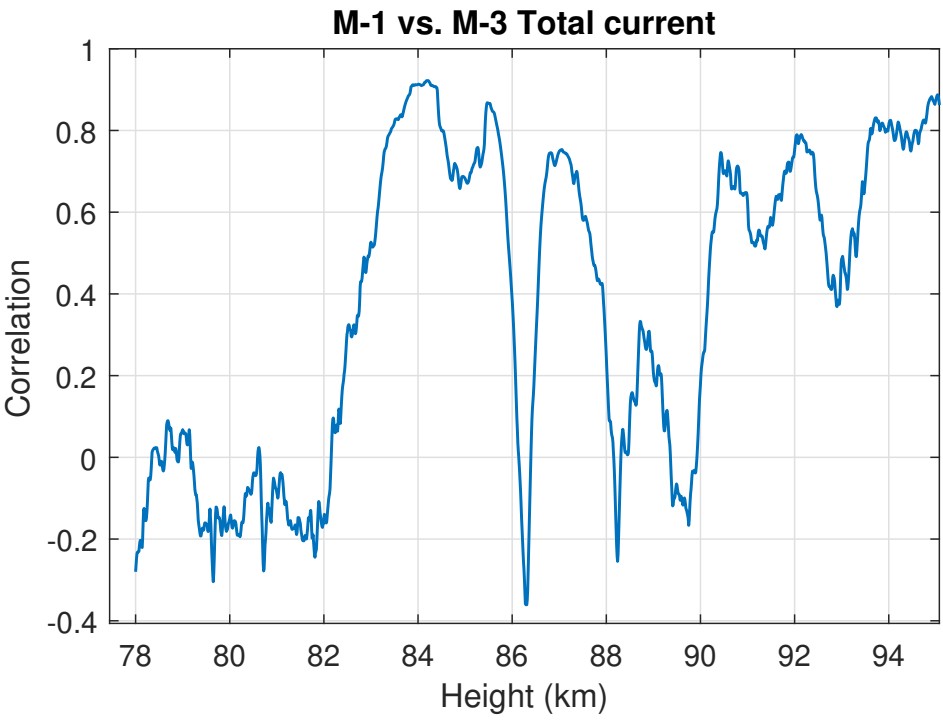

**Figure 9.** Correlation between the total current channels ($U_R = -2$ V) on MUDD-1 and MUDD-3 evaluated for a moving window of 1000 samples corresponding to $\sim 100$ meters in altitude.

## 4.2 Electron density measurements

We must also address the electron population. In a number of studies, a large scale bite-out comparable to the largest dust structure scales have been observed. From earlier studies on small scale correlation between aerosols and electrons it has been found that density variations should follow the same general anti-correlation. However, in some cases, there can be an anti-correlation due to high evaporating rates and other proposed mechanisms (Rapp et al., 2003a; Lie-Svendsen et al., 2003).

There were two instruments measuring electron density on the MXD payloads, by Faraday rotation and needle Langmuir probes (mNLP; see Jacobsen et al. (2010); Bekkeng et al. (2010)). The mNLP electron density is estimated by linearly fitting the electron current at three to four different points in the electron saturation and retardation regions, i.e. without voltage sweeping. Some uncertainty is introduced in regions where OML theory is not completely valid, and the error can be on the order of $10\%$ (Hoang et al., 2018). However, due to its high sampling frequency, and thus ability to resolve relative fluctuations, the m-NLP instrument is much more convenient to compare with DUSTY currents, even though uncertainties may occur due to changes in the floating potential and aerodynamic effects (Private communication, Friedrich, Torkar and Spicher, 2018). For aboslute value comparisons, the Faraday rotation experiment from TU Graz yield accurate absolute electron densities with a lower height resolution. Capacitance probes were also employed to aid in determining suitable observation frequecies for the Faraday Rotation experiment. The Faraday measurements, when compared to the positive ion probe data and charge number density from DUSTY, suggest that the mNLP overestimated the electron density in the cloud layer by a factor of $\sim 10$ for both flights if assuming quasi neutrality. In Fig. 10 we show the comparison of the electron density derived from the UiO mNLP-instrument, using three probes on boom 2 biased at 4.5 V, 6 V and 7.5 V repectively, and DUSTY raw current throughout the entire cloud region. Since particles are predominantly negatively charged, a positive correlation between the curves means a negative correlation between aerosols and electrons. Somewhat surprising, the large scale correlation between electron density and DUSTY current is not as unambiguously positive as expected, but a clear bite-out is present. The variation of the correlation on the largest scales ($\sim 0.1 - 1$ km) are discussed in more detail below. If we look at the correlation, thus anti-correlation between $N_e$ and $N_d$, on scales of length $\sim 10$ m, we see a high similarity between the DUSTY and mNLP curves more or less throughout the dust cloud. In Fig. 11, we show the situation in a $\sim 200$ m thick slice around 84 km. The correlation is close to unity down to scales of a few metres. This should confirm that dust particles are dictating electron dynamics and lower their diffusivity. Since the PMSE during MXD-1B was particularly strong, the scattering structures are probably associated with very steep electron density gradients. A deep look into turbulence and diffusivity of the species will not be done here, but may further corroborate that small particles are in fact accountable for the disagreement between DUSTY-1 and DUSTY-2 currents in parts of the cloud system, as opposed to pure payload potential and aerodynamic adverse flow effects of larger particles. In figure 12 we present the correlations between electron density and DUSTY currents at three different characteristic length scales, corresponding to moving windows of $\sim 10$, 100 and 1000 m. In this calculation, a correlation between electron density and DUSTY currents implies – here as earlier – an anti-correlation between the electron and aerosol population. This is well demonstrated in figure 11, where the curves following each other closely implies that there is almost a one-to-one anti-correlation between electron and aerosol densities. This indicates that the dominating electron loss mechanism is attachment

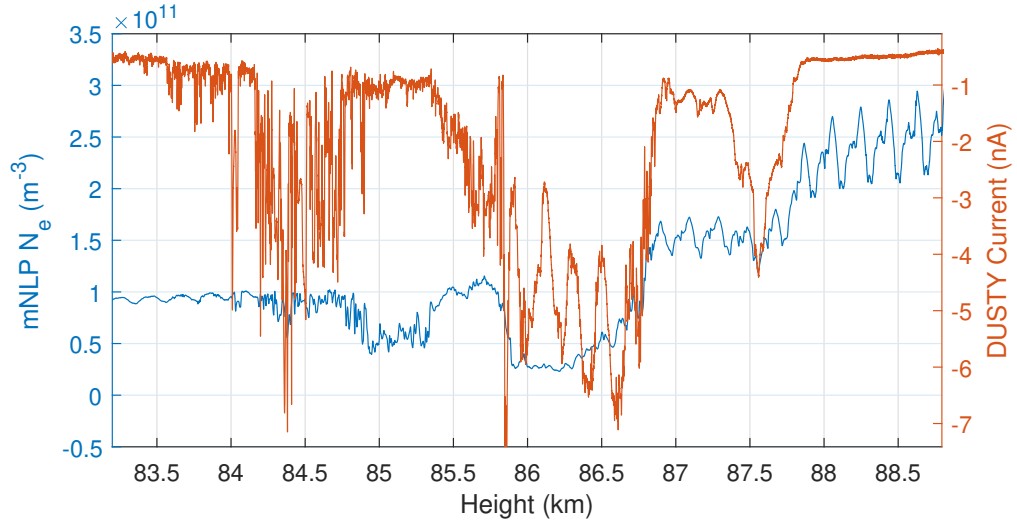

**Figure 10.** Comparison of electron density measure by the mNLP probes (blue) and DUSTY bottom plate current (red).

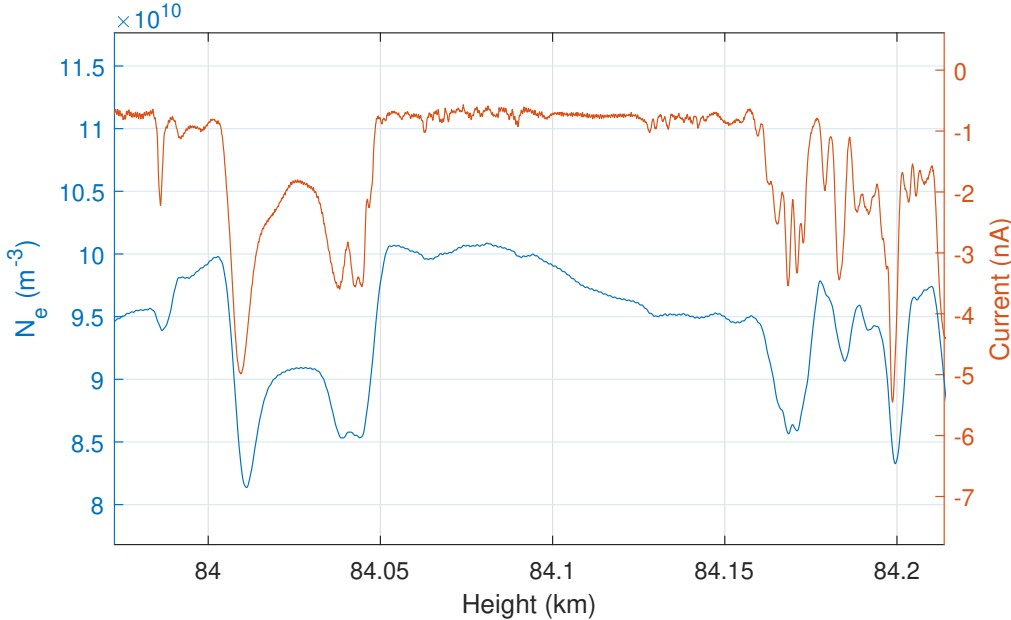

**Figure 11.** Close-up of structure where the electron density and DUSTY-1 currents agree well, during the MXD-1B launch. We note that the electron density height vector is shifted according to the angle between DUSTY-1 and mNLP Boom-1 ($\sim 20$ m in height). We note a correlation on length scales $\sim 10$ m implying anti-correlation between absolute densities.

to aerosols. The curves expectedly show a high degree of similarity, however, by changing the window size we aim to reveal

large scale effects which are otherwise masked by small to mid-scale fluctuations. The overall correlation between electron density and DUSTY current is clearly positive – implying anti-correlation between the densities. With increasing window size, it becomes evident that in the region around $\sim 85.5$ km, where the gradient in the aerosol density and to a certain degree also electron density are steep and the DUSTY currents do not match, the correlation between electron density and the aerosol population becomes positive. This is noteworthy, as a mechanism in which this would happen is difficult to construct. Lie-Svendsen et al. (2003) and Rapp et al. (2003a) points out that a possible positive correlation between dusty plasma species densities could happen if the particles are particularly large with high evaporation rates. As shown in figure 7, the particle sizes are small throughout the cloud system here, so this latter mechanism might be difficult to reconcile with our data. As a last note on the correlation, we look at the situation at $\sim 86.25$ km. This is where the iteration scheme yields the lowest sizes throughout the layer, and it is in the middle of the most active region where the two DUSTY probes show a strong spin modulation. At this point, there is a small region of relatively strong positive correlation between the species densities. A possible effect might be that parts of the payload created a spray of smaller ice particles with a high production of secondary electrons. This may be consistent with one of the capacitance probes mounted on a boom on MXD-1B recording peculiar signals and furthermore that the floating payload potential increases in this region (Private Communication, M. Friedrich). It is also clear that wake effects should play a role, i.e. booms entering and exiting the wake periodically will influence the measurements. The degree to which such wake effects will affect the electron-dust coupling is not however simple to estimate, and future works doing similar studies as the one presented here should consider three-dimensional modelling of the flow field around the payload as two-dimensional axisymmetric flows (see e.g. Antonsen and Havnes (2015)) is insufficient in predicting the aerodynamic environment geometries. A full treatment would probably require Monte Carlo simulations to obtain a statistical mean as the flow is partly rarefied in the cloud region. Calculation of recombination rates, evaporation rates and flow modelling can also be done to give a definitive answer to the question about the observed positive correlation.

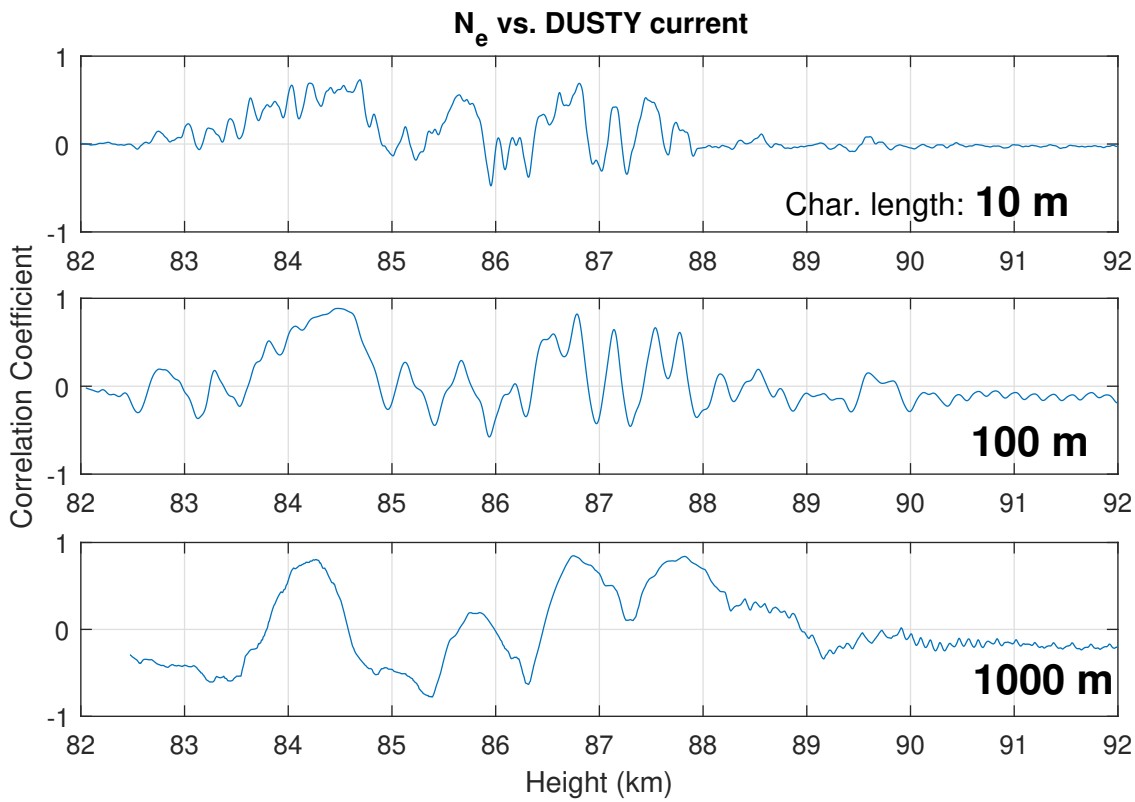

**Figure 12.** Comparison of correlation between needle Langmuir-probe electron density and DUSTY-2B currents. The correlation coefficients are Spearman Rank values for moving windows of three different characteristic lengths; 10, 100 and 1000 m. The 10- and 100 m components were further filtered to remove spin components.

## 4.3 Spectral properties

The connection between the mesospheric aerosol population(s) and PMSE strength, can be characterized through the spectral properties of the cloud sysytem. To assess the spectral properties we utilize wavelet analysis to compute power spectra of the DUSTY currents, as wavelets are much more robust than Windowed Fourier Transforms (WFT) with respect to unwanted features induced by the length of the signal; wavelet transforms conserve both high time and frequency resolution, while in WFT the window length introduces a trade-off between time and frequency resolution. The wavelet transform (WT) is determined theoretically through a convolution between a wavelet and the raw probe current (see e.g. Torrence and Compo (1998)):

$$(I_{BP} * \Psi_\Omega)(\xi) = W_n(s) = \sum_{k=0}^{N-1} I_{BP,k} \Psi_\Omega^* \left[ \frac{(k-n)\delta\xi}{s} \right] \tag{4}$$

where $I_{BP}$ is the DUSTY bottomplate current, $\Psi_\Omega$ is the wavelet for a non-dimensional frequency denoting the number of voices per octave, $\delta\xi$ is the sampling time increment and $s$ the wavelet temporal scales. In the following, we have used the complex Morlet wavelet

$$\Psi_\Omega(\xi) = \Gamma e^{-\xi^2/2} e^{i\Omega\xi} \tag{5}$$

for normalization constant $\Gamma$ and a number of voices per octave of $\Omega = 16$. Similar wavelet transforms have been used by e.g. Brattli et al. (2006), Strelnikov et al. (2009) and Asmus et al. (2017) for spectral analysis of rocket probe data. To obtain the power spectral density (PSD) of the DUSTY signal, we calculate $|W(s)|^2 = WW^*$, which is arbitrarily normalized.

Figure 13 shows a comparison of DUSTY-1B currents, PMSE recorded by the MAARSY radar (IAP Kühlungsborn) along the rocket trajectory and the PSD from wavelet transform in the height region of the dust cloud system during the MXD-1B launch. A striking feature is the strength of the PMSE which peaks at $\sim 50$ dB. The currents recorded by DUSTY-1B are also relatively strong compared to earlier flights. In general the three main 'bumps' in the DUSTY-1B current agree well in altitude with the regions of strongest PSD. The PMSE strength shows no clear agreement with any single feature of the DUSTY signal, but we must note that the PSD strength at wavelengths close to the radar Bragg-scale ($\approx 2.8$ m) is sufficient to be consistent with PMSE throughout the entire region between $\sim 82.5$ and $\sim 86$ km. That is, at these altitudes, the PSD have not reached the steep spectral slope consistent with the viscous convective subrange. In figure 14 we take a closer look at the global power spectrum for the MXD-1B cloud region. Firstly, we note that there is a clear 'knee' between the Kolmogorov and Bachelor subranges ($k \sim -5/3$ and $-1$) and viscous subranges. An interesting observation is that the knee is situated at wavenumbers close to, or even larger than, the radar Bragg scale. This might be expected due to the consistently very strong PMSE throughout the cloud region and may indicate that there is active turbulence, if the aerosols are viewed as passive tracers for the neutrals (see e.g Rapp and Lübken (2004); Driscoll and Kennedy (1985)). A noteworthy feature related to the spectral slope above 85 km should be addressed; When looking at PSD at single heights above this point, it becomes evident that the decay of the curves are in fact generally steeper than what is expected for turbulent layers, and thus edge effects become important (Alcala et al., 2001; Alcala and Kelley, 2001). The implication of this to PMSE proxies is discussed in section 5.

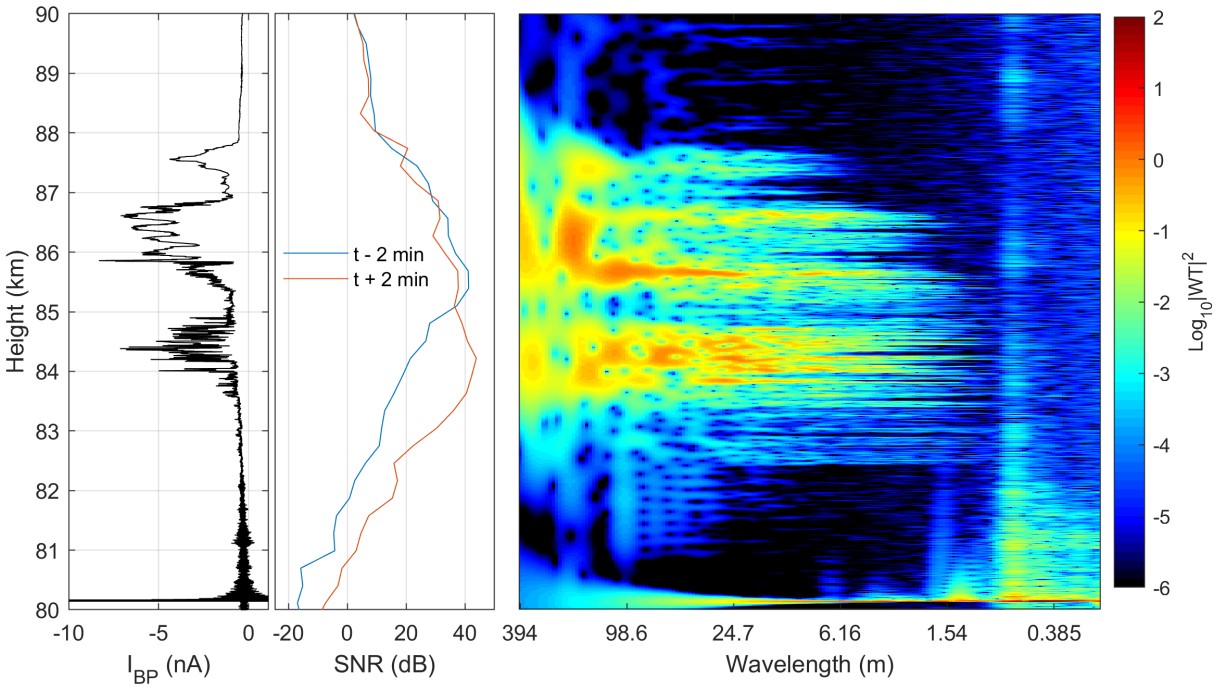

**Figure 13.** Comparison of DUSTY-1B bottomplate current (left panel), MAARSY 53.5 MHZ radar SNR along the rocket trajectory (middle panel) and PSD from wavelet transform (right panel) – for the MXD-1B launch. The spatial scales in the right panel were converted from frequency to approximate wavelength through $\lambda = 2\pi v_R/\omega$, where the rocket velocity was set to that of the middle of the dust cloud; $v_R = 800 \ \mathrm{ms}^{-1}$. Radar data courtesy of Ralph Latteck, IAP Kühlungsborn.

The sharp peak in DUSTY current at just above 80 km is due to a squib firing, and is found to induce noise in a number of harmonics at wavelengths shorter than a few metres in the power spectrum. That the features at these wavelength can be traced to mechanical vibrations induced by a squib firing is confirmed by the power spectrum from the MXD-1 flight, shown in figure 15, where the squib firing at $\sim 83.5$ km produces very similar (transient) noise and harmonics. The noise at short wavelengths below the squib firing can be traced to nosecone separation. The apparent wavelength of the oscillations induced by squib firings are worthwhile discussing. Due to their proximity in wavelength to the radar Bragg-scale – both for the VHF and UHF regime – some caution should be taken when comparing PMSE and PSD. Some harmonics, e.g. at $\sim 0.5$ m in figure 13 are only slowly decaying. Moreover, there seems to be another component modulating the slowly decaying oscillations which in some cases might suggest that such feature is in fact real (which is not the conclusion here). A region of particular interest for the MXD-1B flight is that at the lower edge of the cloud system, between $\sim 82.5$ and $\sim 83.5$ km. In this region, the dust currents are very weak, but there is still significant strength in the PSD, even at wavelengths down to some tens of cm. It is difficult to conclude whether or not UHF PMSE would be observable for these conditions, due to the noise induced by the squib firing. Nevertheless, as is confirmed by the density and radius calculations presented above, there should be a

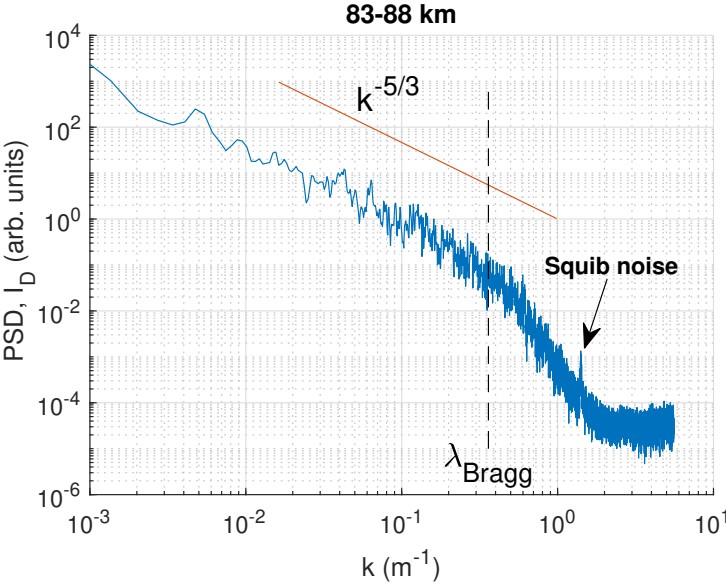

**Figure 14.** Global Power Spectral Density of the cloud region from 83 to 88 km, computed with the Welch method. The Kolmogorov spectral slope is indicated in red. The dashed line indicated the radar Bragg-scale; note that we use linear instead of angular wavenumbers. The squib noise can also be seen in figure 13 as a consistent band thorughout the layer.

small population of large ice particles present in this region which can sustain turbulent structures at short length scales. This may be another reason to expect UHF PMSE more often at the lower edge of of the dust system. The fact that the VHF PMSE is strong in this region, and furthermore stays relatively stationary over a four minute time window around launch, is another confirmation of the presence of particles lowering electron diffusivity. One key observation from the PMSE case during the MXD-1B launch is that even though the VHF PMSE was extremely strong, it does not necessarily imply that the probability for UHF PMSE is high.

For comparison, we present in figure 15 the analogous plot to figure 13 for the MXD-1 flight. We note that the spatial scales indicated for the power spectrum are similar, but we have included a slightly wider range for the MXD-1 flight clearly see the spin noise and its harmonics. The spin components are especially pronounced at wavelengths between $\sim 200$ m and $\sim 20$ m, and the dominant wavelength is consistent with the recorded spin period of 3.7 Hz. There are significant differences between the overall spectral properties of the respective flights. In the MXD-1B flight, the recorded currents and power spectral densities are much stronger in general, compared to the first flight. We note that a strong dust charge number density does not necessarily imply a strong PSD by causality. Similar to the MXD-1B flight, there is a significant strength in the PSD at the lower edge of the cloud system, however we cannot trace the PSD down to scales of tens of cm, due to the noise induced by mechanical vibrations. One feature worth noting, is that it seems that the PSD in general extends down to shorter length scales at lower altitudes, however not significantly stronger in value than expected.

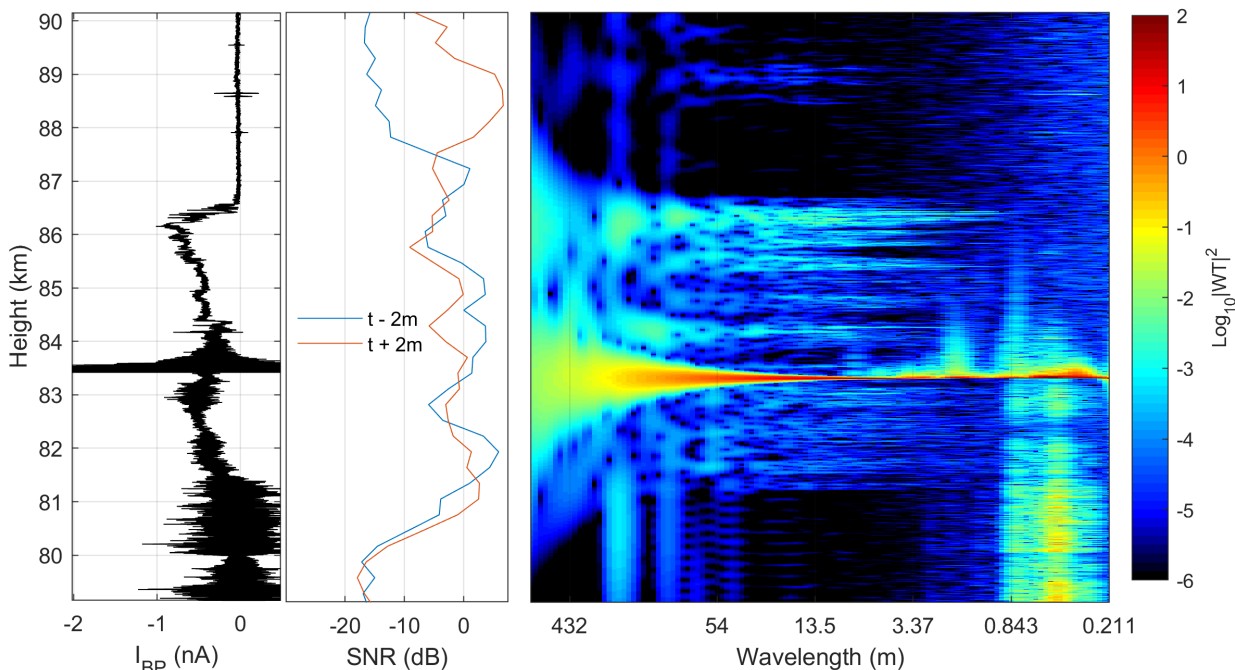

**Figure 15.** Comparison of DUSTY bottom plate current (left panel), MAARSY 53.5 MHZ radar SNR along the rocket trajectory (middle panel) and PSD from wavelet transform (right panel) – for the MXD-1 launch on the 30th of June, 2016. Conversion from frequency to spatial scales is done as in figure 13, by using the mean rocket velocity though out the dust cloud. Radar data courtesy of Ralph Latteck, IAP Kühlungsborn

### 4.4 Aerosol Dependence in PMSE Reflectivity and Proxies

The radar reflectivity in PMSEs have been subjected to much scrutiny since the first observation of coherent VHF echoes, and the exact scattering mechanism is still not agreed upon. However, there is consensus that for relatively low dust concentrations – as falls out from the application of the theory on scattering from Bragg-scales structures in a dusty plasma – that the main part of PMSE modulation must be dependent on the square of the co-dependent dust/electron density gradient (see e.g. Rapp et al. (2008); Varney et al. (2011)) according to:

$$\eta \propto \bar{S}^2 \nabla \langle N_d \rangle^2 \equiv \left( \frac{Z_d N_e}{N_e + Z_i^2 N_i} \right)^2 \cdot \left( \frac{\omega_B^2 N_d}{g} - \frac{\mathrm{d}N_d}{\mathrm{d}t} - \frac{N_d}{H_n} \right)^2 \tag{6}$$

where $\bar{S}/Z_d$ is the mean number of Debye-sphere electrons and $\nabla \langle N_d \rangle$ is the gradient of dust density across an active cloud layer. In the gradient term, $\omega_B$ is the buoyancy frequency, $g$ is the gravitational constant and $H_n$ is the neutral scale height. The full expression for the reflectivity, as provided for the electron-aerosol dusty plasma in the mentioned works, includes a number of ordering parameters, such as the Richardson- and Prandtl-number, as well as microphysical parameters such

as the Batchelor-scale, buoyancy frequency and more. A quick application of the expression is complicated and impractical. Due to this fact, a few ordering parameters and proxies have been suggested as central for the existence of PMSE. The most fundamental dust plasma ordering parameter is the ratio of dust charge number density to electron density, $\Lambda = |N_d Z_d|/N_e$. As pointed out by Bellan (2010), if PMSE is purely from spatial modulation of gas phase electrons due to aerosols, the reflectivity

would scale as some power of $\Lambda/(1+\Lambda)^2$. A few other authors have proposed proxies for PMSE; Rapp et al. (2003b) and Blix et al. (2003) found $N_d|Z_d|r_d^2$ to a consistent proxy for the fossil and active turbulence mechanism of PMSE, while Havnes et al. (2001) did a comparison to $|N_d Z_d|$ – all works with reasonable agreement between proxies and PMSE strength. Furthermore, Havnes (2004) uses the ordering parameter $P \sim N_d r_d/N_e$ for a time dependent cloud model for a Boltzmann distributed plasma, which has been used to predict over- and undershoots of PMSE.

During the campaign, the MAARSY MST radar (see e.g. Latteck et al. (2012)) was run with 2 minute integration time with height bins of $\approx 300$ m. The radar used four beams at inclinations of 0, 8, 12 and 16 degrees from vertical respectively, with an azimuth equal to the rocket launcher settings. The 12 degree beam contained the rocket trajectory in the cloud region and the horizontal sampling region at a given altitude is a circle of approximate area $\approx 3.1$ km$^2$. We must note that even though the rocket trajectory overlaps with a sampled volume, the radar volume is much larger than the volume traversed by the rocket,

and a direct comparison must thus be done with care. However, since MAARSY is able to resolve PMSE three-dimensionally such that SNR can be obtained for azimuths relatively close, a comparison of spectral properties as obtained by DUSTY and PMSE strength can still done on scales on the same order as the vertical resolution of the radar. It is nevertheless clear that a thorough study of edge effects is complicated by the resolution constraints.

In figure 16 we show the comparison of the four key proxies introduced above to PMSE for the MXD-1 flight. The rea-

son why we use the first flight for comparison is due the extraordinary strength and lack of fine structures in the MXD-1B PMSE, thus a comparison with the moderate strength and dynamic situation during the first flight is better suited for proxy comparison. There is a weak total positive correlation for all proxies. It should be noted that none of the proxies predict the reduction in PMSE strength at $\sim 83$ and $85.5$ km well, and the upper and lower edges of the cloud system are poorly represented by all parameters. In a general comparison of proxies, we computed the correlation between all proxies on the form

$\log_{10}(N_d^i|Z_d|^j r_d^k/N_e^l)$ with PMSE SNR, for $\{i,j,k,l\}$ running from 0 to 4. No single proxy scored significantly higher than others, but all proxies in figure 16 were among the highest scoring with correlation coefficients $\lesssim 0.2$. From this simple analysis it is not possible to conclude about the PMSE mechanism, however, it is reasonable to assume that a gradient term should be included.

In the same manner as Rapp et al. (2003b), we look at the relationship between PMSE SNR and $|N_d Z_d|$ in figure 17. In

their figure 10, a pronounced slope of $\sim 1$ supported the validity of a proxy with linear dependence on the dust charge number density. This is not the case for the MXD-1B, where an unambiguous slope cannot be derived.

As a last point of attack in our inquiry into the aerosol/PMSE relationship, we compare in figure 18 the wavelet PSD at a wavelength of 2.8 m, equal to the MAARSY Bragg-scale, to the PMSE SNR throughout the layer for both MXD flights. If the PMSE mechanism was purely from aerosols dictating gas phase electrons, the SNR and PSD would follow each other

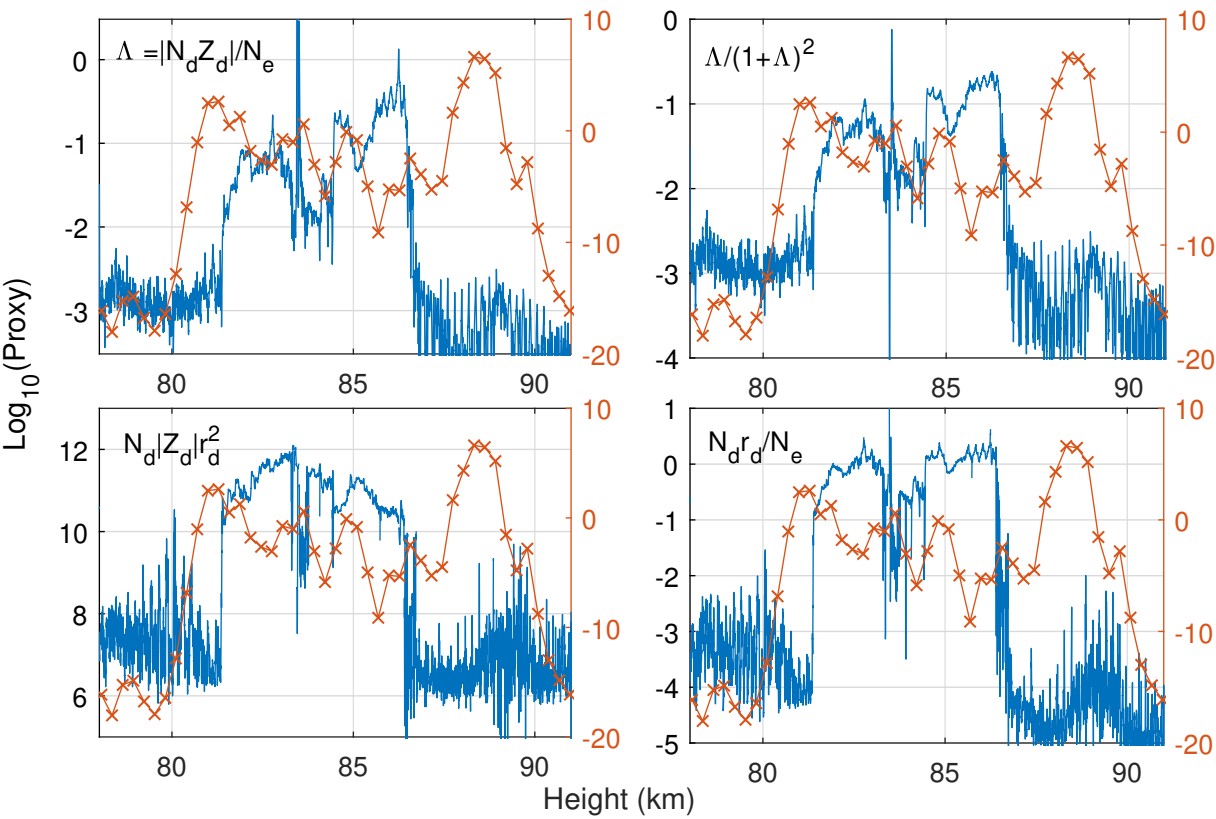

**Figure 16.** Comparison of proxies from dusty plasma parameters to PMSE SNR for the MXD-1 flight. The upper two panels are proxies based on the ratio of dust charge number density to electron density. The proxy in the lower left panel can be recognized as the parameter utilized by Rapp et al. (2003b), while the bottom right panel is the P-factor introduced in Havnes (2004) as an ordering parameter in dust cloud modelling.

closely. Although the PMSE SNR does not display the strong reductions in strength as the PSD, the curves correlate fairly well non-linearly. Again the agreement is low at the edges.

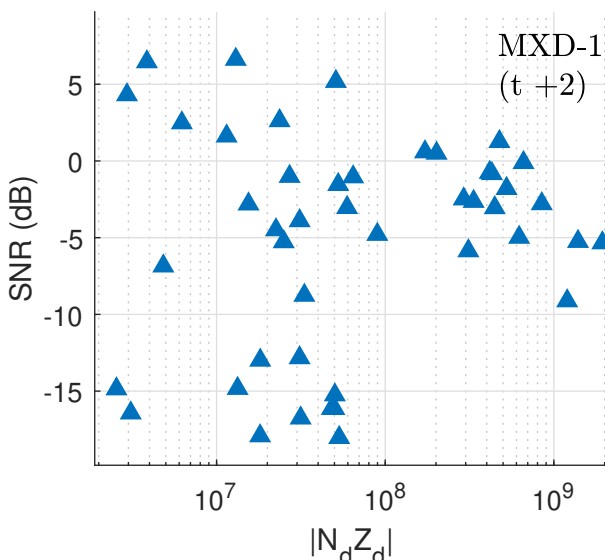

**Figure 17.** Scatter plot of Charge number density derived from DUSTY and PMSE SNR in decibels for the MXD-1 flight. Note that the SNR scale has a range of three orders of magnitude, thus a one-to-one correlation would yield a line with $\lesssim 45°$ angle in this plot. No principal axis can be derived significantly for this cluster, i.e. the variance is too large.

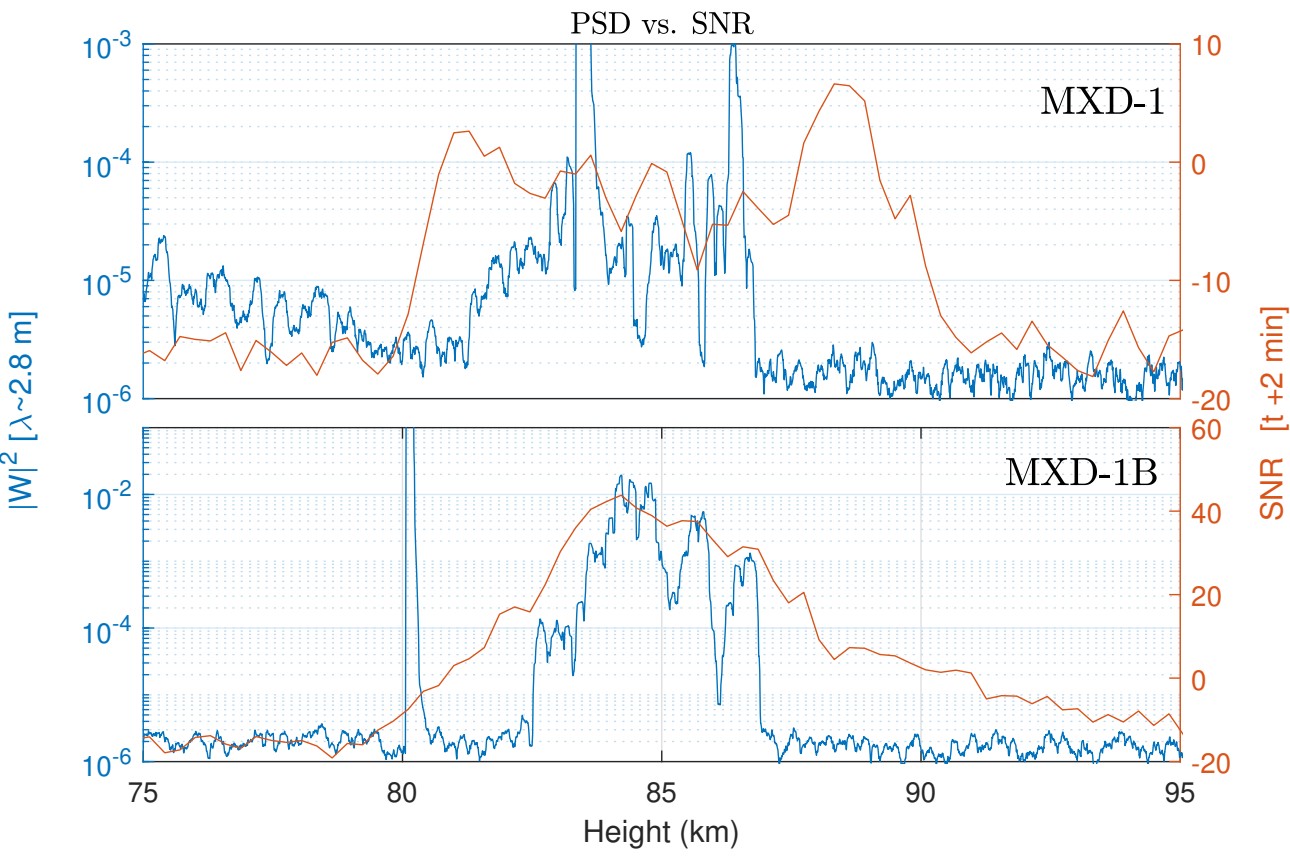

**Figure 18.** Comparison of MAARSY SNR (red, 2 minutes after launch), and PSD from the wavelet transform of DUSTY bottom plate currents (blue), evaluated at a wavelength of $\approx 2.8$ m $= \lambda_{\mathrm{Bragg}}$. The top and bottom panels show the situation for the MXD-1 and MXD-1B launches respectively. Note the different scales for the two panels.

## 5 Discussion

The recorded currents during MXD-1B with large spin modulation and large differences in neighbouring DUSTY and MUDD probes, have two plausible explanations; adverse effects from payload charging with resulting electron leakage or small particles combined with strong aerodynamic modulation. From preliminary estimation of the floating potential from the m-NLP assuming that the probes were in the saturation region, we find that the payload floating potential is only offset with about 3 V on average in the dust cloud region, which would not be enough to let 2-3 eV electrons into DUSTY or MUDD. Another possibility is the presence of very small particles, possibly MSPs, with high enough fraction of the dust charge density to affect the BP currents significantly. From modelling studies it has shown MSPs smaller than $\sim 1 - 2$ nm are swept away or heavily influenced by the neutral flow field in the shock front of the payload (Hedin et al., 2007; Antonsen and Havnes, 2015). In the summer mesopause, the density of MSPs of sizes larger than this cut-off is found to be relatively low in modelling studies, so an in depth analysis of the dynamics of small dust particles around the MXD-1B payload must be carried out. Small particles/MSPs have a rapid density diffusion which implies a rapid smoothing of dust clumps/holes. Particles of sizes $\sim 1 - 2$ nm generally have a charging time much longer than $L/v_R$ (where $L$ is a characteristic length of the payload), so they have the time to spatially modulate electrons even after they enter the shock of the payload without producing a bite-out – or anti-correlation in the respective densities. The last candidate mentioned in this paper as a possible candidate for the strong modulation in DUSTY currents, is the adverse effect of a spray of fragments and secondary charges from a stuck boom above the top deck. We do not address this issue in this paper as it requires substantial three-dimensional modelling of flow around the payload which is not the focus of the present work.

The combination of different perspectives on small scale measurements of mesospheric aerosols and electrons in this work, underlines especially one thing: aerodynamic effects can completely dominate recorded signals in the presence of aerosols. In missions where a relatively high resolution of particle sizes cannot be inferred, particular caution must be taken when analyzing small scale dust phenomena.

In our comparison of the DUSTY currents from MXD-1B with auxiliary measurements of electrons with needle Langmuir probes and dust with the MUDD probe, we find that the agreement is good below a height of $\sim 85.5$ km. Above this, the agreement on shorter scale is less pronounced, however, a large scale bite-out is present. This is to say that all instruments were affected by the same modulation at spin frequency. Interestingly, the electron data displayed little rotational modulation in the layer where DUSTY showed a strong spin component. The explanation of this boils down to the same situation as mentioned above, where aerosols cannot absorb electron quickly enough; this is plausible as the electron attachment rate for both pure ice and MSP particles with sizes below 10 nm is much larger than the time is takes for a particle to traverse the distance from the front of the rocket to the top deck. A more rigorous calculation of electron attachment rates may reveal possible combinations of parameters which produce more effective recombination rates, but generally with $N_e \sim 10^8 - 10^{11}$ m$^{-3}$, the attachment rates for particles $\lesssim 10$ nm are on the order of seconds to hundreds of seconds.

If the aerodynamic environment in front of the payload can be characterized properly, the dual-probe configuration of DUSTY on MXD-1B can also be used to investigate the horizontal differences in small scale dust structures. In the case of

MXD-1B, the interpretation of the data from the region with the strong spin modulation, a possible interpretation could be that there are highly elongated structures consisting of small dust particles which persist in the cloud system for relatively long times. To confirm this, and give a detailed description of the multi-scale structures in the cloud, a rigorous treatment of the dust and electron gradients – in both the vertical and horizontal direction – must be carried out. A work using the two DUSTY probes on MXD-1B to infer aerosol holes and blobs at small scales is currently ongoing.

We must also mention the modest inquiry into the comparison between PMSE and aerosol fluctuations. Generally, the power spectra from fluctuations in the DUSTY currents – directly connected to the aerosol charge number density – agree well *inside* the cloud at the radar Bragg scale, for both flights. How edge effects are manifested in the aerosol fluctuation spectra have not to our knowledge been thoroughly investigated earlier. In addition, a straight forward comparison between PMSE and DUSTY currents give similar conclusions: PMSE edges cannot be described easily from aerosol measurements. Moreover, as MAXIDUSTY is one of few flights where 'all' the relevant dusty plasma parameters are either measured or can be inferred from measurements, we made a comparison of simple proxies for PMSE strength. In this context it may be noted, as found by Alcala et al. (2001); Alcala and Kelley (2001), that for power spectra steeper than the -5/3 slope of Kolmogorov-scale dominated systems, cloud edges dominate the PSD. Consequently, if such steep gradients are seen, it is plausible that a cloud potential model as the one used in Havnes (2004) is the most descriptive for the cloud structures, as edges may be better described from electrostatic effects and Boltzmann distributed plasma species. Regarding a PMSE proxy, this means that the parameter $N_d r_d / N_e$ would be a good ordering parameter, as it is the principal ordering parameter in the mentioned cloud potential model. However, this is not clear in our measurements, as is also the case for the remaining calculated proxies.

## 6  Conclusions

The key findings are summarized as follows:

1. The measurements from two mechanically and electrically identical DUSTY Faraday cups with an interspacing of $\sim 10$ cm show very different measurements in parts of a cloud system (MXD-1B flight). We attribute this to the precence of small particles of sizes $\sim$ a few nanometres which are heavily modulated in the complex aerodynamic environment around the rocket payload.

2. A correlation analysis between charged aerosols and electrons shows very strong negative correlation coefficients on vertical scales of lengths down to $\sim 10$ metres. In a few smaller regions of the dust cloud system, we find weak to medium strong positive correlation between the two species. This effect is difficult to reconcile with the earlier proposed mechanism that the aerosols in this case must be large with a significant evaporation rate. In fact, in the parts of the cloud where positive correlation is seen, the particle sizes are only a few nanometres large.

3. The difference in wavelet power spectra between the MXD-1B flight, where the PMSE was very strong, and the MXD-1 flight, where the PMSE was weak, is significant. For MXD-1B, the PSD keeps its strength to shorter wavelengths

compared to MXD-1. There does not seem to be a clear tendency that a strong spectral power in the VHF regime (on MAARSY with Bragg scales of 2.8 m) implies that the PSD keeps its strength down to the UHF length scales.

4. We find a generally weak agreement between simple proxies from dusty plasma parameters and recorded PMSE strength. Edge effects cannot be reproduced with the proxies or PSD extracted through wavelet analysis at the radar Bragg scale presented in this paper.

*Acknowledgements.* We thank Ralph Latteck at IAP Kühlungsborn for the MAARSY radar profiles for the MAXIDUSTY Campaign, and kind contributions from the University of Oslo by the 4DSpace strategic research initiative. The rocket campaign and the construction of the rocket instrumentation was supported by grants from the Norwegian Space Centre (VIT.04.14.7, VIT.02.14.1, VIT.03.15.7, VIT.03.16.7) and the Research Council of Norway, grant 240065. The replication data for the figures in this paper can be found through the UiT Open Research repository at https://doi.org/10.18710/N8GF1U.

*Competing interests.* The authors declare that they have no conflicts of interest.

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
