# Peer review of "Multi-scale Measurements of Mesospheric Aerosols and Electrons During the MAXIDUSTY Campaign"

_Atmospheric Measurement Techniques, 2018_

## Referee Comment (RC1) · Anonymous Referee #2 · 15 Nov 2018

A plot of global wavelet spectrum ($\sim$100 m height bin) of e.g. the DUSTY-1B would be of great interest. PMSE are thought to be formed by neutral turbulence acting on a dusty plasma. There are spectral models describing the behavior of tracer in such systems (e.g. Driscoll & Kennedy, "Model for the Spectrum of Passive Scalars in an Isotropic Turbulence Field.", Phys. Fluids, 1985). Whether or not the presented results support this should be discussed and will gain the impact of the paper. As radar backscatter in the mesosphere is solely determined by electron density spectral analysis of the electron density measurements would be of interest. Can they reproduce PMSE edges as seen by radar?

[Figure]

Please also note the supplement to this comment:
https://www.atmos-meas-tech-discuss.net/amt-2018-291/amt-2018-291-RC1-supplement.pdf

**Supplement:**

**Referee Report on amt-2018-291**

Title: Multi-scale Measurements of Mesospheric Aerosols and Electrons During the MAXIDUSTY Campaign
Authors: T. Antonsen, O. Havnes and A. Spicher

Content: This paper investigate rocket-borne, multi-scale measurements of mesospheric aerosols and electrons. The measurements are compared to radar echoes of polar mesospheric summer echoes. The main findings are that there is a generally good agreement between spectral power of dust density fluctuations at the radar Bragg scale and the measured radar SNR. The authors state that the edge regions of PMSE are not well correlated with the rocket-borne measurements. In addition, from in-situ measurements derived proxies do not correlate with radar signal strength.

**Comments to the authors:**

A plot of global wavelet spectrum (~100 m height bin) of e.g. the DUSTY-1B would be of great interest. PMSE are thought to be formed by neutral turbulence acting on a dusty plasma. There are spectral models describing the behavior of tracer in such systems (e.g. Driscoll & Kennedy, *"Model for the Spectrum of Passive Scalars in an Isotropic Turbulence Field."*, Phys. Fluids, 1985). Whether or not the presented results support this should be discussed and will gain the impact of the paper.

As radar backscatter in the mesosphere is solely determined by electron density spectral analysis of the electron density measurements would be of interest. Can they reproduce PMSE edges as seen by radar?

---

## Referee Comment (RC2) · Anonymous Referee #1 · 29 Nov 2018

This manuscript is about the interpretation of rocket-borne charged particle measurements by Faraday cup detectors and related plasma instruments. The authors go into great detail discussing the signals detected on different electrodes of the instruments. I regard the resulting step-by-step interpretation of the detected signals as sound. However, it does not make this paper easy to read. There is more focus on understanding what the instruments are seeing than on drawing actual geophysical conclusions. This is in line with a number of earlier publication about the interpretation of similar charged particle measurements from sounding rockets. It is certainly important to understand this kind of particle measurements in more detail. However, from a reader's perspective one would hope that at some point focus would shift towards the "bigger picture",

i.e. the geophysical conclusions that can be drawn from these measurements.

I therefore very much would like to encourage the authors to clearly state in the paper: What are the geophysical research objectives that motivate this paper? What are the new findings of this paper, as compared to earlier publications in the field of charged particle detection? How does this paper bring us closer to drawing conclusions about the geophysics of mesospheric ice and smoke particles? As an example, I would like to point out the last paragraph of the introduction (starting from page 2, line 28). This paragraph very much reads like a Results or Summary section. I would rather like to see that the authors in this part of the introduction clearly state the geophysical questions to be addressed by these rockets flights and by the sets of instruments. Later in the paper, the authors should then return to these questions and state what answers have actually been found ("closure").

As part of providing a "bigger picture", I would like to see several points to be discussed more deeply:

- Several identical detectors (2x DUSTY, 3x MUDD) are flown on each of the two rocket payloads. Comparing their measurements, an important conclusion is that aerodynamic effects are important for the rocket-borne measurement of (small) particles. This is not a new conclusion. Is it possible to make use of having several identical detectors: Can this be used to correct for the aerodynamic effects? Or can this be used to obtain concrete geophysical conclusions? In particular, I wonder about the goal of measuring variability on very short horizontal scales (comparable to the distance of the detectors on the front deck of the payload), as stated e.g. on page 2, lines 30-34. Can this goal be achieved, or is this made impossible by the aerodynamic effects? Much of the discussion in the remainder of the manuscript seems to indicate that the use of several identical detectors does not help us to overcome the aerodynamic obstacles or to arrive at new geophysical conclusions.

- While a major focus of the paper is on understanding the detector signals, still only a

subset of all available measurements (2 payloads carrying 2x DUSTY, 3x MUDD and various plasma probes) is discussed. It would be good to add some statements about those detector signals that are not discussed explicitly. Are they consistent with the major findings of the paper, or are there more aspects?

- A number of issues are not answered as they are "beyond the scope of the current paper". Examples are found ion page 16, line 9-13, page 25, line 15-18, page 25, lines 27-3, and page 26, line 3-4. What is the way forward here? What kind of additional studies would be needed (or are possibly planned)?

I have one other major concern about the paper: A major point of the paper is a detailed comparison of in-situ rocket measurements and simultaneous radar measurements from the ground. In general, the discussion of the various PMSE proxies in section 4.4 is very instructive. However, in order to draw conclusions, it is essential to discuss the actual overlap of the two measurements. Considering the differences in measurement volumes is essential e.g. for the discussion of the PMSE proxies. Given that the radar measurements, is it really possible to draw detailed conclusions about how well different PMSE proxies (based on the rocket measurements) describe the edges of the PMSE region (based on the radar measurements)? The manuscript mentions that the radar data are obtained "along the rocket trajectory" and with an integration time of 2 minutes, but nothing is said about the actual size of the measurement volume over which the radar averages at the altitude of the rocket measurements. Please add this information and a discussion on how this affects the conclusions.

Minor comments:

Abstract: Mention in one of the first sentences that this paper is about sounding rocket experiments.

page 1, line 3: remove comma after "10 cm"

page 1, line 5: When using the word "anti-correlated", make clear: anti-correlated to what?

page 1, line 14: Considering the size range of smoke particles in the mesosphere, I suggest to replace "from nanometer sized" with ""from sub-nanometer sized".

page 1, line 13: What is meant by "lowers the nucleation threshold"? Threshold in terms of what? A nucleation threshold is usually expressed in terms of temperature. In that sense, this sentence's statement that "low temperature lowers the nucleation threshold" does not really make sense.

page 1, line 19: make clear that this refers to the mesopause region "in summer"

page 2, line 1: Add a comma after "balloons".

page 2, line 1: "... rocket probes are..."

page 2, line15: It is not good style to have a headline numbered 1.1 when there is no 1.2.

page 2, line 28: add space before "MAXIDUSTY"

page 1, line 30: Change the format of the citation. Instead of "(see (Havnes et al., 1996))", write "(see Havnes et al. (1996))" or better simply "(Havnes et al., 1996)". Check also the remainder of the text for similar issues with the citation format!

page 2, line 32: define "very short length scales"

page 3, line 29: Some explanation should be added: Why can secondary charge production on G1 (and G0) be neglected when secondary charge production on G2 is a dominant process?

page 3, line 29: "is" should be "it"

Section 3: Information about the payload attitude is central to this discussion. Information about the rocket spin rate ( $\sim$ 3.8 Hz) and about the angle of attack should be

provided earlier in this section. Also the mentioning of "precession" is confusing: The fact that there is an angle of attack is independent of the question whether there is precession or not. However, if there is precession, the angle of attack will vary periodically over time. Is there such a variation of the angle of attack because of precession? This would be important information for the interpretation of the dat. Please clarify this.

Section 3: The radius determination from the charged particle measurements is central to this discussion. This refers to the manuscript Havnes et al. in this special issue. Is there a reference to an accepted paper by now? If not, the basic ideas behind this size analysis method should be re-stated in the current paper.

Figure 7: Are the large particle sizes shown around the edges of the PMSE layer (z<84 km and z>88 km) real? Or is this an artefact of the size analysis method. What are the uncertainties of this size analysis (error bars) as a function of altitude?

page 7, line 18: remove "ratio between"

Section 4: While the design of the DUSTY detector has been described in detail in this paper, can you provide a reference about the design of the MUDD detector?

page 11, line 22: "reveals" instead of "reveal"

page 15, line10: "This, of course meaning..." is not good grammar and should be changed.

page 19, line 30: "accordingly" should be replaced e.g. by "according to"

page 25, line 2: It is not clear what is meant by "yield a large spread" and "horizontal gradients". Please clarify. Does this refer to variations on the scale of the rocket diameter?

page 25, line 27: "which" should be "where"

page 25, line 30: "... from the front of the payload to the top deck." Clarify: do you mean between the shock front and the top deck?

page 26, line 24: "show" should be "shows"

page 26, line 32: "it" should be "its"

page 26, line 31-32: This sentence is unclear and probably grammatically wrong. Please rewrite.

---

## Author Comment (AC1) · 17 Dec 2018

The leftmost panel of Figure 13 displays the global wavelet spectrum power density; it is the wavelet transform multiplied with its complex conjugate and thus displays a power spectral density. We are uncertain about whether or not the global wavelet spectrum can reveal more useful information, as the PSD is the quantity which is more easily relatable to PMSE.

To look at the connection between wavelet spectrum and radar backscatter we use the PSD at wavelengths close to the radar wavelength (+/- 1 wavenumber bin). The result, as shown in Figure 17, shows that edge effects are poorly represented in the PSD, and

that the PSD curves show more structure (due to integration height of PMSE?).

As was pointed out earlier by the referee, it might be very interesting to look at snippets of interesting height regions of the cloud system and discuss the wave number dependency. This is a good point which we intend to include in a revised manuscript. As an example, see the appended figure. This is a PSD (DUSTY data) from a ~400 m slice during MXD-1B. Even though it seems 'all' spectral strength has dissipated at radar wavelengths, the PMSE is still very strong. We note that the raw current is used here, and the true PSD values are several orders of magnitude higher. For MXD-1B, the PMSE SNR is particularly strong in the entire region, and we can find many examples where the spectral strength is low, even though the SNR is strong. For the revised manuscript, we shall carefully consider if a discussion connected to these 'issues' can improve the paper.

As a last point, we need to address the spectral properties of the electrons. As it turns out, the electrons and charged aerosols are well coupled down to the smallest scales. An example is the comparison in Figure 11. Thus, in a spectral analysis, the results will be very similar. Due to this we have omitted an additional wavelet analysis of the electrons here. Similar findings have been reported on earlier. E.g. Rapp, Lübken and Blix ACP, (3)1399-1407 (2003) show in their Figure 7 comparisons of electron and aerosol PSD in short height regions. The spectral slopes are virtually identical over the entire range. Moreover, one must use assumptions to acquire the electron density from the mNLP-probes (and the validity of the theory is not always easy to test for at these heights) while the dust charge number density is correlated one-to-one to the recorded currents.
* * *
**83.85-84.25 km**

Log [(Δ N/N)² /m⁻¹]

$k^{-1}$

$k^{-3}$

$k^{-7}$

m⁻¹

**Fig. 1.**

---

## Author Comment (AC2) · 18 Dec 2018

The authors thank the referee for very useful comments.

The authors can agree with the referee's impression that the geophysical conclusions may be difficult to draw out from the manuscript at this point. We took, however, as a starting point that the type of paper that the present one falls under in most cases benefit from a thorough introduction of instruments and methods, etc. Along the way some important physical discussion may become tangled up in such complicated introductions and important conclusions can become lost. The point is a highly valid one, and we aim to refine the paper in such a way that the main conclusions is not lost to the

reader. The are two main findings of the paper. (1) That there can be large differences on the smallest horizontal and vertical scales, and (2) That even though aerosols and electrons are co-dependent to a high degree at "all" length scales, there is no obvious way to connect it to PMSE strength although such has been proposed earlier.

A consequence of finding (1) is that instruments on the same topdeck on a payload can record very different signals. This is not discussed in the majority of papers treating sounding rocket observation of aerosols. It also confirms what we already know; that small aerosols are strongly dictated by the shock front. We think that this is especially important when identifying small structures in a mesospheric cloud system. It is possible to use two identical probes to correct for aerodynamic effects, but it may be more interesting to use such configurations to analyse the shapes of small structures. We are in fact currently working on a manuscript which compares signals from identical probes and it seems much information can be drawn from such an analysis.

Maybe especially the introduction of the MUDD instrument in the paper draws away from the main findings, and that the manuscript can benefit from removing the MUDD-section? The MUDD measurements are presented to confirm the DUSTY measurements.

We do not in detail discuss the DUSTY and MUDD signals from MAXIDUSTY-1 in this paper. This is because the signals were very similar (large particles dominated the charge number density of aerosols). MXD-1B had a very interesting cloud structure, and the probe currents reflect this. The risk of including MXD-1 is that the paper gets even more involved with discussions that draws away from the big picture (?). We included MXD-1 in the spectral analysis.

The referee points out a very important issue when it comes to simultaneous rocket-radar: The difference in observed volumes. We will add a paragraph on this. For this online comment, we hope it is sufficient to state that the majority (almost all) reflected power in the height region 80-90 km is from circular slices of 1 km diameter. We will

double check all these parameters and add a proper part about them in the manuscript.

The minor comments will be corrected. Some comments that need clarification are answered below:

Page 1: "... lowers the nucleation threshold". This is a poor sentence; maybe it is not necessary to mention threshold at all since it comprises both temperature, saturation, etc.

Page 3: Why can secondary charge production on G1 (and G0) be neglected when secondary charge production on G2 is a dominant process? The effective area of G1 and G0 is negligible compared to G2. The correction to the calculated number charge density is only a couple of percent.

Section 3. Precession. We can neglect that the angle of attack changes significantly over the cloud layer. The precession period is much longer than the spin (O(10) seconds, will add the exact period).

Regarding the 'companion paper' and Figure 7 etc.: The Havnes et al. paper on size inference is in final stage review. If it is not accepted before submission of the present paper, a deeper explanation of the method can be added.
* * *

---

## Author Response (AR1)

**Revised Version 1 of the Manuscript**

«Multi-scale Measurements of Mesospheric Aerosols and Electrons During the MAXIDUSTY Campaign»

We thank the referees for very useful comments. Below is an overview of changes done in the revised version. Minor typographical errors pointed out by the referees are not listed here but have been corrected.

**Changes due to comments from referee #1**

The current paper is focussed around measurements, and it has become evident that the "big picture" is somewhat lost in the current version of the manuscript, as pointed out by the referee. The investigations made in the current work is motivated by the need of a better understanding of what structures (if any) are typical on length scales down to ~10 cm. This is the same as a typical UHF PMSE Bragg-scale, and would be interesting for that reason among others. Also, as the MAXIDUSTY (MXD) payload arguably had one of the more complete setups for investigating aerosols and electrons simultaneously, it seems natural to include a discussion of the correlation between these species, as relatively few works have done the same. One of the main goals behind this is to better understand PMSEs, but is also a good demonstration of experimental capabilities to study blobs and holes in the dusty plasma; which may be useful for a number of inquiries into ordered structure around the mesopause. We also point out that the effect of the aerodynamic environment on the measurement is an interesting by-product of our studies, and that special care needs to be taken in any in-situ probing of this height region. We have altered/added a paragraph on p.2/l.28 – p.3/l.13 that aims to communicate the points above in a better manner than in the original manuscript.

***On aerodynamic effects***: For the MUDD probe, having three probes offers a few advantages; more probes means a faster sampling/sweeping of the 12 potential modes (corresponding to mass bins of meteoric smoke/fragments). This means that we can, in the case of MXD,  get a three times higher altitude resolution of the size distribution of meteoric smoke embedded in ice, which is desired. As the probe share one potential mode with another probe, one can in principle correct for differences due to aerodynamic effects (however, in Antonsen et al (2017) the size distribution was calculated in regions which were fairly homogeneous and calm). Also, as is the motivation behind having two identical DUSTY probes, it offers the possibility to look at small horizontal scales. As large ~10 nm particles are not influenced by the aerodynamic shock front (as discussed in e.g. Horanyi et al. (1999), Hedin et al (2007), Antonsen and Havnes (2015)) it should be possible to infer structures at these small horizontal scales – if they exist. To our knowledge, there are no studies discussing observations of such short horizontal scales, and MXD is one of few payloads for mesosphere studies that facilitates for such measurements. The conclusions to draw from finding differences on short scales is perhaps a bit difficult to communicate in a short paragraph, but we hope that inferring holes and 'blobs' of charged aerosols on these small scales can be used in studies of e.g. UHF PMSE. In this paper we feel it is important to underline that large differences on these scales are not necessarily true variations in the dusty plasma, but probably due to small particles. This conclusion would be more difficult to come to if we did not employ more than one dust detector. We feel the justification in Section 3 Paragraph 2 and discussions in the following paragraphs communicates the points above well.

***On other available measurements***: Other probes that have not been discussed are Positive Ion Probes, Capacitance probes, the miniMASS dust mass spectrometer and the ICON neutral mass spec. - The ICON data is currently being studied, and understanding the data is arguably more difficult than understanding the normal plasma and dust probes. As the first results are being presented only later this year, we do not want to include ICON in the manuscript.

- miniMASS measurements were corrupted by photoelectrons as an unfortunate geometry and insufficient shielding couldn't prevent sunlight to hit the detector. No dust could be inferred unambiguously.

- The Daughter Payload module was a proof of concept and did not produce publishable data (the first data from a regular flight [G-Chaser] is being presented later this year).

- The capacitance probe on MXD-1 functioned well, but only above ~84 km due to a relatively low electron density. For MXD-1B, one of the probes give useful data from ~76 km, which confirms that the electron density measurements by Faraday rotation are sound; i.e. what frequency will give the best result (added a note on this in section 4.2). One of the capacitance (high freq) probes did not however work nominally on MXD-1B, and according to M. Friedrich, it may indicate that the boom did not employ correctly (specified this in 4.2, Private Comm. Friedrich).

- The positive ion probes worked nicely for both flights, and give densities which are consistent with the calculated electron densities, however, indicate that the electron density from mNLP is probably slightly overestimated in the cloud region. Added a sentence on this on p.14/l.15.

***On topics 'beyond the scope…'***: We have done only a low resolution 3D-simulation of the aerodynamic environment which is not sufficient to predict the MXD flow patterns. A thorough 3D-simulation, probably using Monte Carlo statistics as the flow is more or less rarefied below ~90km, is demanding and is thus not addressed. Added a sentence on this p.16/l.17-21. Regarding p.26/l.3-4, we are currently working on a manuscript using the identical DUSTY probes to infer the structure of small holes and blobs in the dusty plasma, but the results are not ready or necessarily well fitted to include in the present discussion. Added sentence on p.26.

***On the concern regarding PMSE***: This is a very good point which needs to be addressed. The strength of MAARSY is that it can resolve PMSE more or less three-dimensionally with a height resolution of ~300 m (see also figure of typical MAARSY plot below). Although some interpolation is done in the horizontal dimensions, we feel that the PMSE measurements should be representable for a region very close to the rocket payload. Edge effects of PMSE and very localized effects in the dusty plasma are of course topics which must be addressed with caution. We have added a new paragraph on p. 20-21 trying to justify our use of the radar, together with technical specifications.

**Changes due to comments from referee #2**

Unfortunately, there is some noise of currently unknown source in the electron current data for MXD-1B. The noise is at wavelengths close to 1.2 m  (see figure 2 below) and affects neighbouring spectral regions as well. Due to this, it is difficult to get a reliable PSD for the shorter scales, and the transition between inertial and viscous subranges are unclear for the electron data.

We have however included a global PSD for the dust data at this point. Although a global wavelet power spectrum is preferable, we have utilized a Welch PSD estimate for the cloud region instead, as it is much less computationally demanding, and offers a very good frequency resolution. We have only

included the global spectrum for MXD-1B, as it had a strong PMSE throughout the entire dust cloud region. The radar Bragg-scale is more or less at the end of the Bachelor/Kolmogorov subrange which may indicate that the turbulent energy dissipation comes from relatively recent/ongoing turbulence. We added a short discussion about this on p. 18. This discussion can also be longer, however, turbulence is not one of the main topics of the current manuscript.

**Other Changes**

Section 4.2, paragraph 2: Added a sentence about how electron density is calculated and possible error. Added ref. Hoang et al.

p.1: Removed 'nucleation threshold'. Added 'allows for nucleation of ice…'

p.3. Secondandy changes from G1 and G0 can from most cases be neglected due to the difference in secondary producing area. The area of G1 and G0 is approximately 18 percent of the area og G2, and since the secondary charge producing part of this is ~28% (Havnes and Næsheim (2007)), the correction would only be a few percent. In the size calculations done in the companion paper we take this into account.

Section 3: There is now only a mention of coning. There is a weak variation due to precession, but throughout the cloud layer (83-88) the angle of attack changes so little, that no significant variation is found).

Section 3: Added about the iteration method. And the artefacts (due to a 1/0 ratio in one of the iterated eqs.)

[Figure]

**Figure 1**: PMSE measurements during MXD-1 showing the horizontal slices.

[Figure]

**Figure 2**: Close-up of 6V NLP current vs. DUSTY currents indicating a strong 644 Hz noise component.

---

## Author Response (AR2)

**Final revision of the Manuscript**

«Multi-scale Measurements of Mesospheric Aerosols and Electrons During the MAXIDUSTY Campaign»

We thank again for the very useful comments of the referees. Below we describe the few changes made to the final revision of the manuscript

**Changes in the final revision**

We have added, in the beginning of section 4, a more complete description of the available data from the Faraday cups (in total 27 channels for the two flights). For the present paper there is an accompanying paper by Havnes et al. in the same issue which describes the DUSTY measurements in more detail. The paper by Antonsen et al (2017) studies the MUDD measurements from the same flight, and we have tried to avoid too large of an overlap. We also added a reference of how to use the same type of Faraday cups for in-flight calibration of electron probes (this is however difficult for MXD since the needle-Langmuir probes require flow regimes valid for OML).

A few typographical errors were also corrected.